# GRAINet: Mapping grain size distributions in river beds from UAV images with convolutional neural networks

**Nico Lang[1], Andrea Irniger[2], Agnieszka Rozniak[1], Roni Hunziker[2], Jan Dirk Wegner[1], and Konrad Schindler[1]**

[1]EcoVision Lab, Photogrammetry and Remote Sensing, ETH Zürich, Switzerland
[2]Hunziker, Zarn & Partner, Aarau, Switzerland

**Correspondence:** Nico Lang (nico.lang@geod.baug.ethz.ch)

**Abstract.** Grain size analysis is the key to understand the sediment dynamics of river systems. We propose *GRAINet*, a data-driven approach to analyze grain size distributions of entire gravel bars based on georeferenced UAV images. A convolutional neural network is trained to regress grain size distributions as well as the characteristic mean diameter from raw images. *GRAINet* allows the holistic analysis of entire gravel bars, resulting in *(i)* high-resolution estimates and maps of the spatial grain size distribution at large scale, and *(ii)* robust grading curves for entire gravel bars. To collect an extensive training dataset of 1,491 samples, we introduce *digital line sampling* as a new annotation strategy. Our evaluation on 25 gravel bars along six different rivers in Switzerland yields high accuracy: The resulting maps of mean diameters have a mean absolute error (MAE) of 1.1 cm, with no bias. Robust grading curves for entire gravel bars can be extracted if representative training data is available. At the gravel bar level the MAE of the predicted mean diameter is even reduced to 0.3 cm, for bars with mean diameters ranging from 1.3 cm to 29.3 cm. Extensive experiments were carried out to study the quality of the digital line samples, the generalization capability of *GRAINet* to new locations, the model performance w.r.t. human labeling noise, the limitations of the current model, and the potential of *GRAINet* to analyze images with low resolutions.

---

## 1 Introduction

Understanding the hydrological and geomorphological processes of rivers is crucial for their sustainable development, so as to mitigate the risk of extreme flood events and to preserve the biodiversity in aquatic habitats. Grain size data of gravel- and cobble-bed streams is key to advance the understanding and modelling of such processes (Bunte and Abt, 2001). The fluvial morphology of the majority of the world's streams is heavily affected by human activity and construction along the river (Grill et al., 2019). Human interventions like gravel extractions, sediment retention basins in the upper catchments, hydro power plants, dams, or channels reduce the bedload and lead to surface armouring, clogging of the bed, and latent erosion (Surian and Rinaldi, 2003; Simon and Rinaldi, 2006; Poeppl et al., 2017; Gregory, 2019). Consequently, the natural alteration of the river bed is hindered, eventually deteriorating habitats and potential spawning grounds. Moreover, the process of bedload transport can cause bed or bank erosion, the destruction of engineering structures (e.g., due to bridge scours) or increased flooding due to deposits in the channel that amplify the impact of severe floods (Badoux et al., 2014). What makes modelling of fluvial morphology challenging are the mutual dependencies between the flow field, grain size, its movement and the geometry of the channel bed and banks. While channel shape and roughness define the flow field, the flow moves sediments — depending on their size — and the bed is altered by erosion and deposition. This mutually reinforcing system makes understanding channel form and process hard. Transport calculations in numerical models are thus still based on empirical formulas (Nelson et al., 2016).

One important key indicator for modelling sediment dynamics of a river system is the *grading curve* of the sediment. Depending on the complexity of the model, the grain size distribution is either described by its characteristic diameters (e.g., the mean diameter $d_m$ defined by Meyer-Peter and Müller, 1948) or by the fractions of the grading curve

(fractional transport, Habersack et al., 2011). The grain size of the river bed is crucial because it defines the roughness of the channel as well as the incipient motion of the sediment (Bunte and Abt, 2001). Thus, knowledge of the grain size distribution is essential to specify flood protection measures, to asses bed stability, to classify aquatic habitats, and to evaluate geological deposits (Habersack et al., 2011). Collecting the required calibration data to describe the composition of a river bed is time-consuming and costly, since it varies strongly along the downstream of a river (Surian, 2002; Bunte and Abt, 2001) and even locally within individual gravel bars (Babej et al., 2016; Rice and Church, 2010). Traditional mechanical sieving to classify sediments (Krumbein et al., 1938; Bunte and Abt, 2001) requires a substantial amount of skilled labour, and the whole process of digging, transport, and sieving is time-consuming, costly, and destructive. Consequently, it is rarely implemented in practice. An alternative way of sampling sediment is surface sampling along transects or on regular grid. We refer to Bunte and Abt (2001) for a detailed overview of traditional sampling strategies. A simplified, efficient approach that collects sparse data samples in the field is the *line sampling* analysis of Fehr (1987), the quasi-gold standard in practice today.[1] This procedure of surface sampling is commonly referred to as pebble counts along transects (Bunte and Abt, 2001). Yet, this approach is still very time-consuming and, worse, potentially inaccurate and subjective (Bunte and Abt, 2001; Detert and Weitbrecht, 2012). Moreover, in-situ data collection requires physical access and cannot adequately sample inaccessible parts of the bed, such as gravel bar islands (Bunte and Abt, 2001).

An obvious idea to accelerate data acquisition is to estimate grain size distribution from images. So-called *photosieving* methods that manually measure gravel sizes from ground level images (Adams, 1979; Ibbeken and Schleyer, 1986) were first proposed in the late 1970s. While the accuracy of measuring the size of individual grains may be compromised compared to field sampling, manual image-based sampling brings many advantages in terms of transparency, reproducibility, and efficiency. Since it is non-destructive, multiple operators can label the exact same location. Much research tried to automatically estimate grain size distributions from ground level images (Butler et al., 2001; Rubin, 2004; Graham et al., 2005; Verdú et al., 2005; Detert and Weitbrecht, 2012; Buscombe, 2013; Spada et al., 2018; Buscombe, 2019; Purinton and Bookhagen, 2019). On the contrary, relatively little research has addressed the automatic mapping of grain sizes from images at larger scale (Carbonneau et al., 2004, 2005; Black et al., 2014; de Haas et al., 2014; Carbonneau et al., 2018; Woodget et al., 2018; Zettler-Mann and Fonstad, 2020), needed for practical impact. Monitoring of river systems over time suffers from biases introduced by different operators in the field (Wohl et al., 1996). Hence, objective, automatic methods for large scale grain size analysis offer great potential for consistent monitoring over time.

Other researchers have proposed to analyze 3D-data acquired with terrestrial or airborne LiDAR, or through photogrammetric stereo matching (Brasington et al., 2012; Vázquez-Tarrío et al., 2017; Wu et al., 2018; Huang et al., 2018). However, working with 3D-data introduces much more overhead in data processing compared to 2D-imagery. Moreover, terrestrial data acquisition lacks flexibility and scalability, while airborne LiDAR remains costly (at least until it can be recorded with consumer-grade UAVs). Photogrammetric 3D-reconstruction is limited by the reduced resolution of the reconstructed point clouds (relative to that of the original images), which suppresses smaller grains. Woodget et al. (2018) have shown that, for small grain sizes, image-based texture analysis is beneficial over roughness-based methods.

While automatic grain size estimation from ground-level images is more efficient than traditional field measurements (Wolman, 1954; Fehr, 1987; Bunte and Abt, 2001), it is commonly less accurate and scaling to large regions is hard. Threshold-based image analysis for explicit gravel detection and measurements is affected by lighting variations and thus requires much manual parameter tuning. In contrast, statistical approaches avoid explicit detection of grains and empirically correlate image content with the grain size measurement. Although these data-driven approaches are promising, their predictive accuracy and generalization to new scenes (e.g., airborne imagery at country-scale) is currently limited by hand-designed features and small training datasets.

In this paper, we propose a novel approach based on convolutional neural networks (CNN) that efficiently maps grain size distributions over entire gravel bars, using georeferenced and orthorectified images acquired with a low-cost UAV. Not only allows our generic approach to estimate the full grain size distribution at each location in the orthophoto, but also to estimate characteristic grain sizes directly using the same model architecture (Fig 1). Since it is hard to collect sufficiently large amounts of labeled training data for hydrological tasks (Shen et al., 2018), we introduce *digital line sampling* as a new, efficient annotation strategy.[2] Our CNN avoids explicit detection of individual objects (grains) and predicts the grain size distribution or derived variables directly from the raw images. This strategy is robust against partial object occlusions and allows for accurate predictions even with coarse image resolution, where the individual small grains are not visible by the naked eye. A common characteristic of most research in this domain is that grain

---

[1]To the best of our knowledge, this includes at least the German-speaking countries: Switzerland, Germany, and Austria.

[2]It is worth noting that the annotation strategy and the CNN are not tightly coupled. Since the CNN is agnostic, it could be trained on grain size data created with different sampling strategies to meet other national standards.

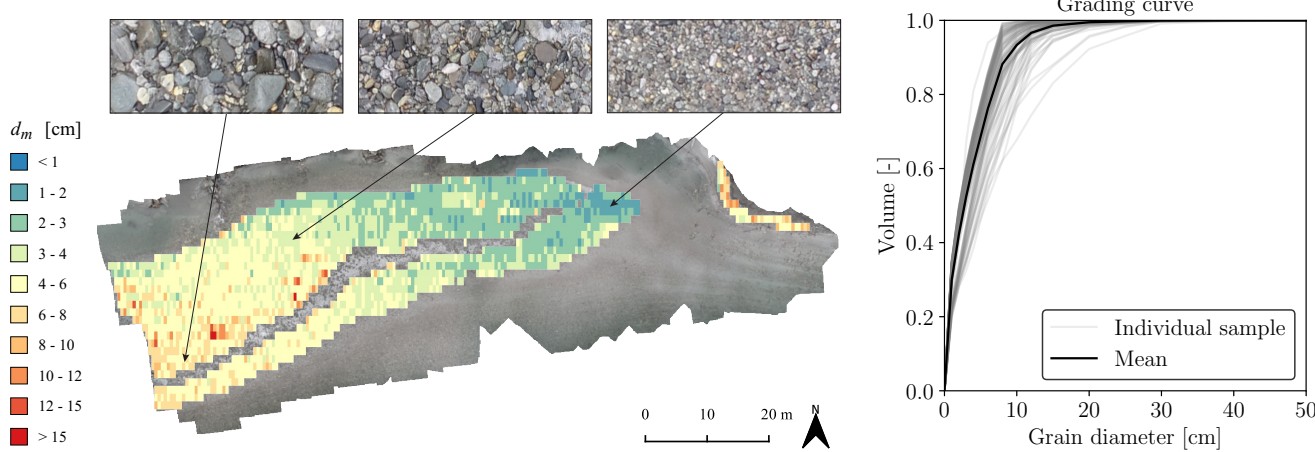

**Figure 1.** Illustration of the two final products generated with *GRAINet* on the river Rhone. Left: Map of the spatial distribution of characteristic grain sizes (here $d_m$). Right: Grading curve for the entire gravel bar population, by averaging the predicted curves of individual line samples.

size is estimated in pixels (Carbonneau et al., 2018). Typically, the image scale is determined by recording a scale bar in each image, which is used to convert the grain size into metric units (e.g., Detert and Weitbrecht, 2012), but limits large-scale application. In contrast, our approach estimates grain sizes directly in metric units from orthorectified and georeferenced UAV images.

We evaluate the performance of our method and its robustness to new, unseen locations with different imaging conditions (e.g., weather, lighting, shadows) and environmental factors (e.g., wet grains, algae covering) through cross-validation on a set of 25 gravel bars. Like Shen et al. (2018), we see great potential of deep learning techniques in hydrology and hope that our research constitutes a further step towards its widespread adoption. To summarize, our presented approach includes the following contributions:

- End-to-end estimation of the full grain size distribution at particular locations in the orthophoto, over areas of $1.25\,\mathrm{m} \times 0.5\,\mathrm{m}$.

- Robust mapping of grain size distribution over entire gravel bars.

- Generic approach to map characteristic grain sizes with the same model architecture.

- Mapping of mean diameters $d_m$ below 1.5 cm.

- Robust estimation of $d_m$, for arbitrary ground sampling distances up to 2 cm.

## 2 Related Work

In this section, we review related work on automated grain size estimation from images. We refer the reader to Piégay

et al. (2019) for a comprehensive overview of remote sensing approaches on rivers and fluvial geomorphology. Previous research can be classified into *traditional image processing* and *statistical approaches*.

*Traditional image processing*, also referred to as object-based approaches (e.g. Carbonneau et al., 2018), has been applied to segment individual grains and measure their sizes, by fitting an ellipse and reporting the length of its minor axis as the grain size (Butler et al., 2001; Sime and Ferguson, 2003; Graham et al., 2005, 2010; Detert and Weitbrecht, 2012; Purinton and Bookhagen, 2019). Detert and Weitbrecht (2012) presented *BASEGRAIN*, a MATLAB-based object detection software tool for granulometric analysis of ground-level, top-view images of fluvial, non-cohesive gravel beds. The gravel segmentation process includes gray-scale thresholding, edge detection, and a watershed transformation. Despite this automated image analysis, extensive manual parameter tuning is often necessary, which hinders the automatic application to large and diverse sets of images. Recently Purinton and Bookhagen (2019) introduced a python tool called *Pebble-Counts* as a successor of *BASEGRAIN* replacing the watershed approach with $k$-means clustering.

*Statistical approaches* aim to overcome limitations of object-centered approaches by relying on global image statistics. Image texture (Carbonneau et al., 2004; Verdú et al., 2005), auto-correlation (Rubin, 2004; Buscombe and Masselink, 2009), wavelet-transformations (Buscombe, 2013), or 2D spectral decomposition (Buscombe et al., 2010) are used to estimate the characteristic grain sizes like the mean ($d_m$) and median ($d_{50}$) grain diameters. Alternatively, one can regress specific percentiles of the grading curve individually (Black et al., 2014; Buscombe, 2013, 2019).

Buscombe (2019) proposed a framework called SediNet, based on CNNs, to estimate grain sizes as well as shapes

from images. Overall, the used dataset of 409 manually labeled sediment images was halved into training and test portions, and CNNs were trained from scratch, despite the small amount of data.[3]

In contrast to previous work, we view the frequency or volume distribution of grain sizes as a probability distribution (of sampling a certain size), and fit our model by minimising the discrepancy between the predicted and ground truth distributions. Our method is inspired by Sharma et al. (2020) who proposed HistoNet to count objects in images (soldier fly larvae and cancer cells) and to predict absolute size distributions of these objects directly, without any explicit object detection. The authors show that end-to-end estimation of object size distributions outperforms baselines using explicit object segmentation (in their case with Mask-RCNN, He et al., 2017). Even though Sharma et al. (2020) avoid explicit instance segmentation, the training process is supervised with a so-called count map derived from a pixel-accurate object mask, which indicates object sizes and locations in the image. In contrast, our approach requires neither a pixel-accurate object mask nor a count map for training, which are both laborious to annotate manually. Instead, the CNN is trained by simply regressing the grain size distribution end-to-end. Labeling of new training data becomes much more efficient, because we no longer need to acquire pixel-accurate object labels. Our model learns to estimate object size frequencies by looking at large image patches, without access to explicit object counts or locations.

## 3    Data

We collected a dataset of 1,491 digitised line samples acquired from a total of 25 different gravel bars on six Swiss rivers (see Table C1 in the appendix for further details). We name gravel bar locations with the river name and the distance from the river mouth in kilometers.[4] All gravel bars are located on the northern side of the Alps, except for two sites at the river Rhone (Fig. 2). All investigated rivers are gravel rivers with gradients of 0.01-1.5 %, with the majority (20 sites) having gradients <1.0 %. The river width at the investigated sites varies between 50 m and 110 m, whereby *Emme km 005.5* and *Emme km 006.5* correspond to the narrowest sites and *Reuss km 017.2* represents the widest one.

One example image tile from each of the 25 sites is shown in Fig. 3. This collection qualitatively highlights the great variety of grain sizes, distributions, and lighting conditions (e.g., shadows, hard and soft light due to different weather

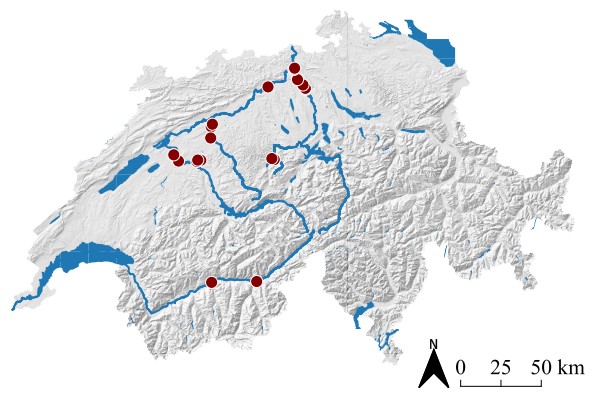

**Figure 2.** Overview map with the 25 ground truth locations of the investigated gravel bars in Switzerland.

conditions). The total amount of digital line samples collected per site varies between 4 (*Reuss km 021.4*) und 212 (*Kl. Emme km 030.3*), depending on the spatial extent and the variability of grain sizes within the gravel bar.

### 3.1    UAV imagery

We acquired images with an off-the-shelf consumer UAV, namely the DJI PHANTOM 4 PRO. Its camera has a 20 Mega-pixel CMOS sensor (5472×3648 pixels) and a nominal focal length of 24 mm (35 mm format equivalent).[5] Flight missions were planned using the flight planner Pix4D capture.[6] Images were taken on a single grid, where adjacent images have an overlap of 80 %. To achieve a ground sampling distance of ≈0.25 cm, the flying height was set to 10 m above the gravel bar. This pixel resolution allows the human annotator to identify individual grains as small as 1 cm. Furthermore, to avoid motion blur in the images, the drone was flown at low speed. We generated georeferenced orthophotos with AgiSoft PhotoScan Professional.[7]

The accuracy of the image scale has a direct effect on the grain size measurement from georeferenced images (Carbonneau et al., 2018). To assure that our digital line samples are not affected by image scale errors, we compare them with corresponding line samples in the field and observe good agreement. Note that absolute georeferencing is not crucial for this study. Because ground truth is directly derived from the orthorectified images, potential absolute georeferencing errors do not affect the processing.

### 3.2    Annotation strategy

We introduce a new annotation strategy (Fig. 4), called *digital line sampling*, to label grain sizes in orthorectified images.

---

[3]While not clearly explained in Buscombe (2019), the results seem to suffer from overfitting, due to a flaw in the experimental setup. Our review of the published source code revealed that the stopping criterion for the training uses the test data, leading to overly optimistic numbers.

[4]With the exception of location  *Emme - ,* a gravel pile outside the channel.

[5]dji.com/ch/phantom-4-pro (2020-03-23)
[6]pix4d.com (2020-04-04)
[7]agisoft.com (2020-04-04)

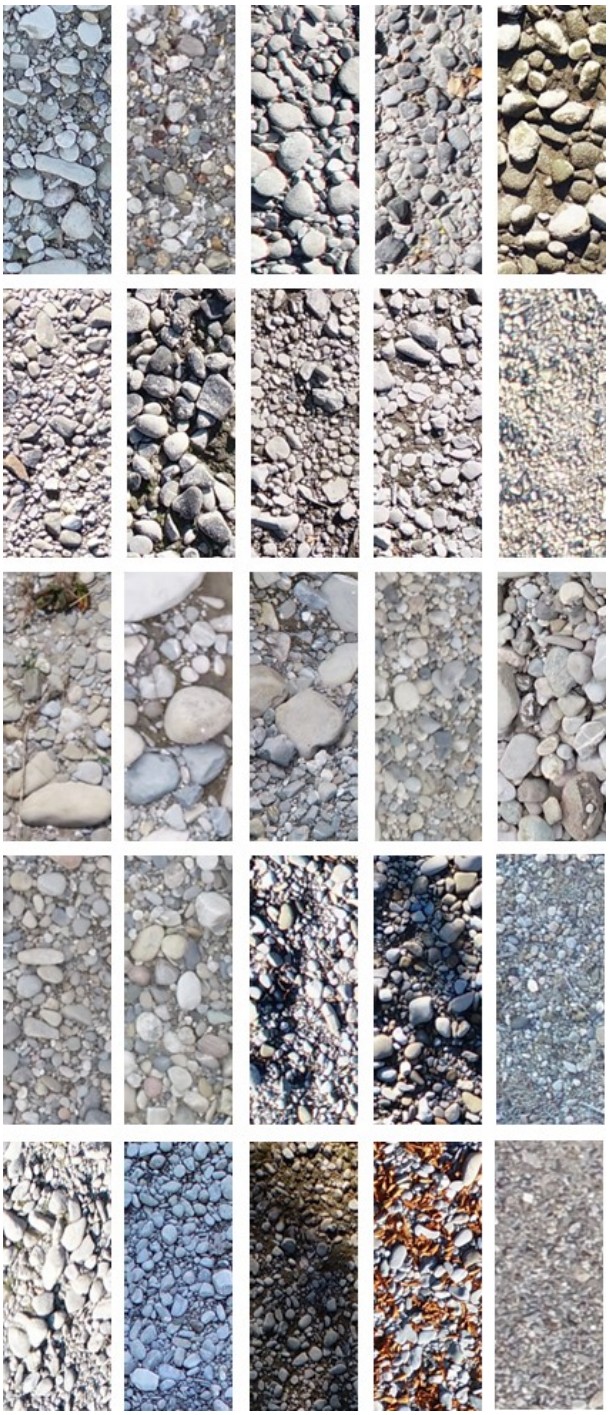

**Figure 3.** Example image tiles (1.25 m × 0.5 m) with 0.25 cm ground sampling distance. Each of the 25 example tiles is taken from a different gravel bar.

To allow for a quick adoption of our proposed approach, we closely follow the popular *line sampling* field method introduced originally by Fehr (1987). Instead of measuring grains in the field, we carry out measurements in images. First, orthorectified images are tiled into rectangular image patches with a fixed size of 1.25 m × 0.5 m. We align the major axis with the major river flow, either north-south or east-west. A human annotator manually draws polygons of 100–150 grains along the center line of a tile (Fig. 4 a) which takes 10-15 minutes per sample on average. We asked annotators to imagine the outline of partially occluded grains if justifiable. Afterwards, the minor axis of all annotated grains is measured by automatically fitting a minimum bounding rectangle around the polygons (Fig. 4 b). Grain sizes are qunatized into 21 bins as shown in Fig. 5, which leads to a relative frequency distribution of grain sizes (Fig. 4 c). Line samples are first converted to a quasi-sieve throughput (Fig. 4 d) by weighting each bin with the weight $w_b = d_{mb}{}^\alpha$ (Fehr, 1987), where $d_{mb}$ is the mean diameter per bin and $\alpha$ is set to 2 (assuming no surface armouring). Usually undersampled finer fractions are predicted by a Fuller distribution, which results in the final *grading curve* (Fig. 4 e). This grading curve can either be directly used for fractional bedload simulations, or to derive characteristic grain sizes corresponding to the percentiles of the grading curve (Fig. 4 f). These are needed, for instance, to calculate the single grain bedload transport capacity ($d_{50}$, $d_{65}$, $d_m$), to determine the flow resistance ($d_m$, $d_{90}$), and to describe the degree of surface armouring ($d_{30}$, $d_{90}$; Habersack et al., 2011).

Our annotation strategy has several advantages. First, digital line sampling is the one-to-one counterpart of the current state-of-the-art in the digital domain. Second, the labeling process is more convenient, as it can be carried out remotely and with arbitrary breaks. Third, image-based line sampling is repeatable and reproducible. Multiple experts can label the exact same location, which makes it possible to compute standard deviations and quantify the uncertainty of the ground truth. Finally, *digital line sampling* allows one to collect vast amount of training data, which is crucial for the performance of CNNs. For modern machine learning techniques, data quantity is often more important than quality, as shown for example in Van Horn et al. (2015). As it is common machine learning terminology, we use the term *ground truth* to refer to the hand-annotated *digital line samples* that are used to train and evaluate our model.

### 3.3 Ground truth

In total, >180,000 grains over a wide range of sizes have been labeled manually (Fig. 5). Individual grain sizes range from 0.5 cm to approx. 40 cm. The major mode of individual grain sizes is between 1 and 2 cm and the minor mode between 4 and 6 cm. Mean diameters $d_m$ per site vary between 1.3 cm (Aare km 178.0) and 29.3 cm (Gr. Entle km 002.0) with a global mean of all 1,491 annotated line sam-

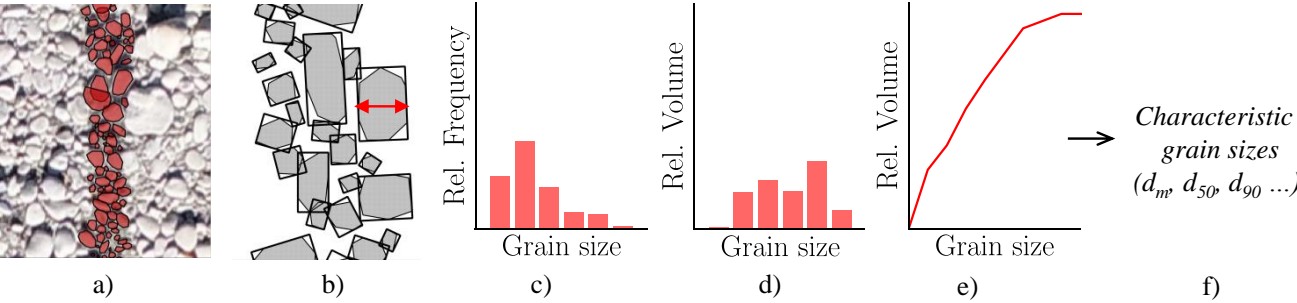

**Figure 4.** Overview of the line sampling procedure. (a) Digital line sample with 100–150 grains, (b) Automatic extraction of the b-axis, (c) Relative *frequency* distribution of grain sizes, (d) Relative *volume* distribution, (e) Grading curve, (f) Characteristic grain sizes (e.g. $d_m$)

ples at 6.2 cm and a global median at 5.3 cm. The distribution of the mean diameters $d_m$ (Fig. 5, right) follows a bi-modal distribution as well. The major mode is around 4 cm and the minor mode around 8 cm. We treat all samples the same and do not further distinguish between shapes when training our CNN model for estimating the size distribution, such that the learned model is universal and applicable to all types of gravel bars. Furthermore, to train a robust CNN, we not only collect easy (clean) samples, but also challenging cases with natural disturbances such as grass, leaves, moss, mud, water, and ice.

## 4   Method

Many hydrological parameters are continuous by nature and can be estimated via regression. Neural networks are generic machine learning algorithms that can perform both classification and regression. In the following, we discuss details of our methodology for regressing grain size distributions of entire gravel bars from UAV images.

### 4.1   Image preprocessing

Before feeding image tiles to the CNN, we apply a few standard pre-processing steps. To simplify the implicit encoding of the metric scale into the CNN output, the ground sampling distance (GSD) of the image tiles is unified to 0.25 cm. The expected resolution of a 1.25 m x 0.5 m tile after the re-sampling is $500 \times 200$ pixels. Inaccuracies may arise due to rounding effects from the prior cropping. For simplicity, the tile size is cropped to $500 \times 200$ pixels. Additionally, horizontal tiles are flipped to be vertical.

Finally, following best practice for neural networks, we normalize the intensities of the RGB channels to be standard normal distributed with mean 0 and standard deviation 1, which leads to faster convergence of gradient-based optimization (LeCun et al., 2012). It is important to note that any statistics used for pre-processing must be computed solely from the training data, and then applied unaltered to the training, validation, and test sets.

### 4.2   Regression of grain size distributions with GRAINet

Our CNN architecure, which we call *GRAINet*, regresses grain size distributions and their characteristic grain sizes directly from UAV imagery. CNNs are generic machine learning algorithms that learn to extract texture and spectral features from raw images to solve a specific image interpretation task. A CNN consists of several convolutional (CONV) layers that apply a set of linear image filter kernels to their input. Each filter transforms the input into a feature map by discrete convolution, i.e., the output is the dot product (scalar product) between the filter values and a sliding window of the inputs. After this linear operation, non-linear *activation functions* are applied element-wise to yield powerful non-linear models. The resulting *activation maps* are forwarded as input to the next layer. In contrast to traditional image processing, the parameters of filter kernels (*weights*) are learned from training data. Each filter kernel ranges over all input channels $f_{in}$ and has a size of $w \times w \times f_{in}$, where $w$ defines the kernel width. While a kernel width of 3 is the minimum width required to learn textural features, $1 \times 1$ filters are also useful to learn the linear combination of activations from the preceding layer.

A popular technique to improve convergence is *batch normalization* (Ioffe and Szegedy, 2015), i.e., re-normalizing the responses within a batch after every layer. Besides better gradient propagation, this also amplifies the non-linearity (e.g. in combination with the standard ReLU activation function).

Our proposed *GRAINet* is based on state-of-the-art residual blocks introduced by He et al. (2016). An illustration of our GRAINet architecture is presented in the appendix in Fig. B1. Every residual block transforms its input using three convolutional layers, each including a batch normalization and a ReLU activation. The first and last convolutional layers consist of $1 \times 1 \times f_{in}$ filters (CONV $1\times1$) while the second

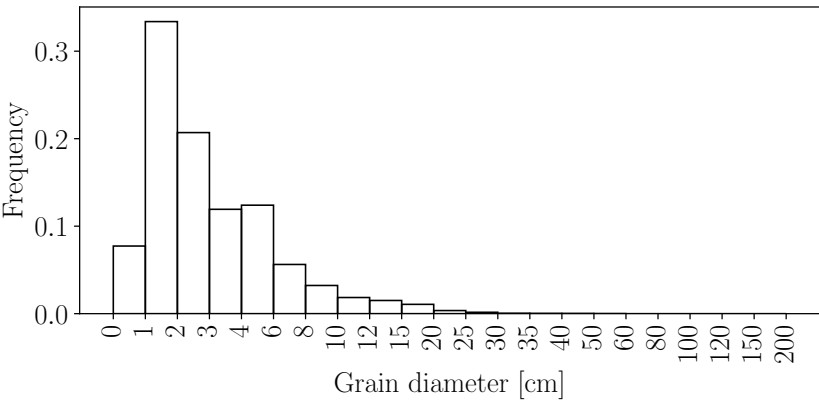 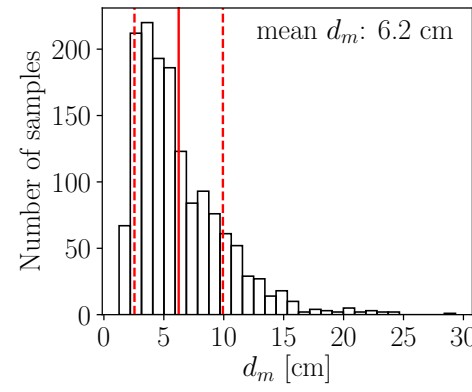

**Figure 5.** Overview of the ground truth data. Left: Average of the 1,491 relative frequency distributions, Right: Histogram of the respective characteristic mean diameter $d_m$. The solid red line corresponds to the mean $d_m$ and the dashed red lines to mean $\pm$ std.

layer has $3 \times 3 \times f_{in}$ filters (CONV 3×3). Beside this series of transformations the input signal is also forwarded through a shortcut, a so called *residual connection* and added to the output of the residual block. This shortcut allows the training signal to propagate better through the network. Every second block has a step size (*stride*) of 2, so as to gradually reduce the spatial resolution of the input image and thereby increase the *receptive field* of the network.

We tested different network depths (i.e. number of blocks / layers) and found the following architecture to work best: *GRAINet* consists of a single 3×3 "entry" CONV layer followed by six residual blocks and a 1×1 CONV layer that generates $B$ final activation maps. These activation maps are reduced to a one-dimensional vector of length $B$ using global average pooling, which computes the average value per activation map. If the final target output is a scalar (i.e. a characteristic grain size like the $d_m$) $B$ is set to 1. To predict a full grain size distribution, $B$ equals the number of bins of the discretized distribution. Finally, the vector is passed through a softmax activation function. The output of that operation can be interpreted as a probability distribution o er grain size bins, since the softmax scales the raw network output such that all vector elements lie in the interval $[0, 1]$ and sum up to one. [8] The total number of parameters of this network architecture is 1.6 million, which is rather lean compared to modern image analysis networks that often have >20 million parameters.

### 4.2.1 CNN output targets

As CNNs are modular learning machines, the same CNN architecture can be used to predict different outputs. As al-

ready described, we can predict either discrete (relative) distributions, or scalars such as a characteristic grain size. We thus train *GRAINet* to directly predict the outputs proposed by Fehr (1987) at intermediate steps (Fig. 4):

(i) relative frequency distribution (*frequency*)

(ii) relative volume distribution (*volume*)

(iii) characteristic mean diameter ($d_m$).

### 4.2.2 Model learning

Depending on the target type (probability distribution or scalar), we choose a suitable *loss* function (i.e., error metric; Sect. 4.3) that is minimized by iteratively updating the trainable network parameters. We initialize network weights randomly and optimize with standard mini-batch stochastic gradient descent (SGD). During each *forward pass* the CNN is applied to a *batch* (subset) of the training samples. Based on these predictions, the difference to ground truth is computed with the loss function, which provides the supervision signal. To know in which direction the weights should be updated, the partial derivative of the loss function is computed w.r.t. every weight in the network. By applying the chain rule for derivatives, this gradient is *back-propagated* through the network from the prediction to the input (*backward pass*). The weights are updated with small steps in negative gradient direction. A hyper-parameter called the *learning rate* controls the step size. In the training process, this procedure is repeated iteratively, drawing random batches from the training data. One training *epoch* is finished once all samples of the training dataset have been fed to the model (at least) once.

We use the ADAM optimizer (Kingma and Ba, 2014) for training, which is a popular adaptive version of standard SGD. ADAM adaptively attenuates high gradients and amplifies low gradients by normalizing the global learning rate with a running average for each trainable parameter. Note

---

[8]In contrast to Sharma et al. (2020) we estimate relative instead of absolute distributions. While they show that the $L^1$-loss and the KL-divergence can be combined to capture scale and shape of the distribution, respectively, we simply fix the scale of the predicted distribution with a softmax before the output.

that SGD acts as a strong regularizer, as the small batches only roughly approximate the true gradient over the full training dataset. This allows training neural networks with millions of parameters.

To enhance the diversity of the training data, many techniques for image data augmentation have been proposed, which simulate natural variations of the data. We employ randomly horizontal and vertical flipping of the input images. This makes the model more robust and, in particular, avoids overfitting to certain sun angles with their associated shadow directions.

### 4.3   Loss functions and error metrics

Various error metrics exist to compare ground truth distributions to predicted distributions. Here, we focus on three popular and intuitive metrics that perform best for our task: the Earth mover's distance (short EMD; also known as the Wasserstein metric), the Kullback-Leibler divergence (KLD), and the Intersection over Union (IoU; also known as the Jaccard index).

The Earth mover's distance (Eq. 1) views two probability density functions (PDF) $p$ and $q$ as two piles of earth with different shapes and describes the minimum amount of "work" that is required to turn one pile into the other. This "work" is measured as the amount of moved earth (probability mass) multiplied by its transported distance. In the one-dimensional case, the Earth mover's distance can be implemented as the integral of absolute error between the two respective cumulative density functions (CDF) $P$ and $Q$ of the distributions (Ramdas et al., 2017). Furthermore, for discrete distributions, the integral simplifies to a sum over $B$ bins.

$$\text{EMD}(P,Q) = \sum_{b=1}^{B} |P(b) - Q(b)| \qquad (1)$$

Alternatively, the Kullback-Leibler divergence (Eq. 2) is widely used in machine learning because minimizing the forward KLD is equivalent to minimizing the negative likelihood or the cross-entropy (up to a constant). It should be noted though that Kullback-Leibler divergence is not symmetric. For a supervised approach the forward KLD is used, where $p$ denotes the true distribution and q the predicted distribution. This error metric only accounts for errors in the bins that actually contain a ground truth probability mass $p(b)$>0. Errors in empty ground truth bins do not contribute to the forward KLD. Therefore, optimizing the forward KLD has a mean-preserving behaviour. In contrast, the reverse KLD is mode-preserving. Note that the KLD only accounts for errors in bins containing ground truth probability mass. Thus, the overestimation of empty bins does not directly contribute to the error metric, but as we treat the grain size distri-

bution as a probability distribution, this displaced probability mass is missing in the bins that are taken into account.[9]

$$D_{\text{KL}}(p \parallel q) = \sum_{b=1}^{B} p(b) \log\left(\frac{p(b)}{q(b)}\right) \qquad (2)$$

In contrast to the EMD and KLD, the Intersection over Union (Eq. 3) is an intuitive error metric that is maximized and ranges between 0 and 1. While it is often used in object detection or semantic segmentation tasks, it allows to compare two 1D-probability distributions as follows:

$$\text{IoU}(p,q) = \frac{\sum_{b=1}^{B} \min(p(b), q(b))}{\sum_{b=1}^{B} \max(p(b), q(b))}. \qquad (3)$$

During the training process the loss function (Eq. 4) simply averages the respective error metric over all samples within a training batch. To evaluate performance, we average the error over the unseen test dataset:

$$\mathcal{L} = \frac{1}{N} \sum_{i=1}^{N} D\left(y_i, f(x_i)\right), \qquad (4)$$

where $D$ corresponds to the error metric, $f$ denotes the CNN model, $N$ the number of samples, $x_i$ the input image tile, $y_i$ the ground truth PDF or CDF, and $f(x_i)$ the predicted distribution, respectively.

To optimize and evaluate CNN variants that directly predict scalar values (like for example GRAINet, which directly predicts the mean diameter $d_m$) we investigate two loss functions: the mean absolute error (MAE, also known as $L^1$-loss, Eq. 5), and the mean squared error (MSE, also known as $L^2$-loss, Eq. 6).

$$\text{MAE} = \frac{1}{N} \sum_{i=1}^{N} |f(x_i) - y_i| \qquad (5)$$

$$\text{MSE} = \frac{1}{N} \sum_{i=1}^{N} \left(f(x_i) - y_i\right)^2. \qquad (6)$$

Furthermore, we evaluate the model bias with the mean error (ME):

$$\text{ME} = \frac{1}{N} \sum_{i=1}^{N} f(x_i) - y_i, \qquad (7)$$

where a positive mean error indicates that the prediction is greater than the ground truth.

---

[9]For completeness, we note that there is a smoothed and symmetric (but less popular) variant of the KLD, the Jensen-Shannon divergence.

## 4.4 Evaluation strategy

The trained *GRAINet* is quantitatively and qualitatively evaluated on a holdout test set, i.e., a portion of the dataset that was not seen during training. We analyze error cases and identify limitations of the proposed approach. Finally, with our image-based annotation strategy multiple experts can label the same sample, which we exploit to relate the model performance to the variation between human expert annotations.

### 4.4.1 Ten-fold cross-validation

To avoid any train-test split bias, we randomly shuffle the full dataset and create 10 disjoint subsets, such that each sample is contained only in a single subset. Each of these subsets is used once as the hold-out test set, while the remaining 9 subsets are used for training *GRAINet*. The validation set is created by randomly holding out 10 % of the training data, and used to monitor model performance during training and to tune hyper-parameters. Results on all 10 folds are combined to report overall model performance.

### 4.4.2 Geographical cross-validation

Whether or not a model is useful in practice strongly depends on its capability to generalize across a wide range of scenes unseen during training. Modern CNNs have millions of parameters and in combination with their non-linear properties, these models have high capacity. Thus, if not properly regularized or if trained on a too small dataset, CNNs can potentially memorize spurious correlations specific to the training locations which would result in poor generalization to unseen data. We are particularly interested if the proposed approach can be applied to a new (unseen) gravel bar. In order to validate if *GRAINet* can generalize to unseen river beds we perform geographical cross-validation. All images of a specific gravel bar are held out in turn and used to test the model trained on the remaining sites.

### 4.4.3 Comparison to human performance

The predictive accuracy of machine learning models depend on the quality of the labels used for training. In fact, label noise that would lead to inferior performance of the model is introduced partially by the labeling method itself. Grain annotation in images is somewhat subjective and thus differs across different annotators. The advantage of our *digital line sampling* approach is that multiple experts can perform the labeling at the exact same location, which is infeasible if done in-situ because *line sampling* is disruptive and cannot be repeated. We perform experiments to answer two questions. First, what is the variation of multiple human annotations? Second, can the CNN learn a proper model, despite the inevitable presence of some label noise? We randomly selected 17 image tiles that are labeled by five skilled operators, who are familiar with traditional *line sampling* in the field.

## 4.5 Final products

On one hand, by combining the output of *GRAINet* trained to either predict the *frequency* or the *volume* distribution with the approach proposed by Fehr (1987), we can obtain the grading curve (cumulative volume distribution) as well as the characteristic grain sizes (e.g., $d_m$). On the other hand, *GRAINet* can also be trained to directly predict characteristic grain sizes. The characteristic grain size $d_m$ is only one example of how the proposed CNN architecture can be adapted to predict specific aggregate parameters. Ultimately, the *GRAINet* architecture allows one to predict grain size distributions or characteristic grain sizes densely for entire gravel bars, with high spatial resolution and at large scale, which makes the (subjective) choice of sampling locations redundant. These predictions can be further used to create two kinds of products, illustrated in Fig. 1:

1. Dense high-resolution maps of the spatial distribution of characteristic grain sizes.

2. Grading curves for entire gravel bars, by averaging the grading curves at individual line samples.

## 4.6 Experimental setup

For all experiments the data is separated into three disjoint sets, a *training* set to learn the model parameters; a *validation* set to tune hyper-parameters and to determine when to stop training to avoid overfitting; and a *test* set used only to assess the performance of the final model.

The initial learning rate is empirically set to 0.0003 and each batch contains 8 image tiles, which is the maximum possible within the 8 GB memory limit of our GPU (Nvidia GTX 1080). While we run all experiments for 150 epochs for convenience, the final model weights are not defined by the last epoch, but taken from the epoch with the lowest validation loss. An individual experiment takes less than 4 hours to train. Due to the extensive cross-validation, we parallelize across multiple GPUs to run the experiments in reasonable time.

## 5 Experimental results

Our proposed *GRAINet* approach is quantitatively evaluated with 1,491 digital line samples collected on orthorectified images from 25 gravel bars located along six rivers in Switzerland (Sect. 3). We first analyze the quality of the collected ground truth data by comparing our digital line samples with field measurements. We then evaluate the performance of *GRAINet* for estimating the three different outputs:

(i) relative frequency distribution (*frequency*)

(ii)  relative volume distribution (*volume*)

(iii)  characteristic mean diameter ($d_m$)

In order to get an empirical upper bound for the achievable accuracy, we compare the performance of *GRAINet* with the variation of repeated manual annotations. All reported results correspond to random 10-fold cross-validation, unless specified otherwise. In addition, we analyze the generalization capability of all three *GRAINet* models with the described geographical cross-validation procedure and investigate the error cases to understand the limitations of the proposed data-driven approach. Finally, as our CNN does not explicitly detect individual grains, we investigate the possibility to estimate grain sizes from lower image resolutions.

## 5.1  Quality of ground truth data

We evaluate the quality of the ground truth data in two ways. First, the *digital line samples* are compared with state-of-the-art in-situ line samples from field measurements. Second, the label uncertainty is studied by comparing repeated annotations by multiple skilled operators.

### 5.1.1  Comparison to field measurements

From 22 out of the 25 gravel bars, two to three field measurements from experienced experts were available (see Fig. 6). These field samples were measured according to the line sampling proposed by Fehr (1987). To compare the digital and in-situ line samples, we derive the $d_m$ values and compare them at the gravel bar level, because the field measurements are only geo-localized to that level. Some field measurements were accomplished a few days apart from the UAV surveys. We expect grain size distributions to remain unchanged, as no significant flood event occurred during that time. Figure 6 indicates that the field-measured $d_m$ is always within the range of the values derived from the *digital line samples*. Furthermore, the mean of the field samples agrees well with the mean of the digital samples. Comparing the mean $d_m$ derived from field and digital line samples across the 22 bars results in a mean absolute error of 0.9 cm and in a mean error (bias) of -0.3 cm, which means that the digital $d_m$ is on average slightly lower than the $d_m$ derived from field samples. The wide range of the *digital line samples* emphasizes that the the choice of a single line sample in the field is very crucial and that it requires a lot of expertise to chose a few locations that yield a meaningful sample of the entire gravel bar population. Considering that the field samples are unavoidably affected by the selected location and also by operator bias (Wohl et al., 1996), we conclude that within reasonable expectations the digital line samples are in good agreement with field samples and constitute representative *ground truth* data. Nevertheless, to better understand the difference between digital line sampling and field sampling, a new dataset should be created in the future, where field samples are precisely geolocated to allow a direct comparison at the tile level.

### 5.1.2  Label uncertainty from repeated annotations

We compute statistics of three to five repeated annotations of 17 randomly selected image tiles (see Table D1 in the appendix) to analyze the (dis-)agreement between human annotators. The standard deviation of $d_m$ across different annotators varies between 0.1 cm (*Aare km 172.2*) and 2.0 cm (*Rhone km 083.3*), the average standard deviation is 0.5 cm. Although these 17 samples are too few to compute reliable statistics, we get an intuition for the uncertainty of the *digital line samples*. Figure 7 shows two different annotations for the same image tile, to demonstrate the variation introduced by the subjective selection of valid grains. While the distribution in the upper annotation (in green) contains a larger fraction of smaller grains following closely the center line, the lower annotation (in blue) contains a larger fraction of larger grains, including some further away from the center line.

Recall that this comparison of multiple annotators is only possible because *digital line sampling* is non-destructive. In contrast, even though variations of similar magnitude are expected in the field, a quantitative analysis is not easily possible. Nevertheless, Wohl et al. (1996) found that sediment samples are biased by the operator. Although CNNs are known to be able to handle a significant amount of label noise if trained on large datasets (Van Horn et al., 2015), the uncertainty of the manual ground truth annotations is also present in the test data and therefore represents a lower bound for the performance of the automated method. Therefore, while we do not expect the label noise to degrade the CNN training process, we do not expect root mean square errors below 0.5 cm due to the apparent label noise in the test data.

## 5.2  Estimation of grain size distributions

As explained in Sect. 3.2, the process of obtaining a grading curve according to Fehr (1987) involves several empirical steps (Fig. 4). In this processing pipeline, the relative frequency distribution can be regarded as the initial measurement. However, as the choice of the proper CNN target is *a priori* not clear, we investigate the two options to estimate *(i)* the relative *frequency* distribution and *(ii)* the relative *volume* distribution. In the latter version, the CNN implicitly learns the conversion from frequency to fraction weighted quasi-sieve throughput, making that processing step obsolete. We experiment with three loss functions to train *GRAINet* for the estimation of discrete target distributions: the Earth mover's distance (EMD), the Kullback-Leibler divergence (KLD), and the Intersection over Union (IoU). For each trained model all three metrics are reported in Table 1. The standard deviation quantifies the performance variations across the 10 random data splits. Theoretically, one would expect

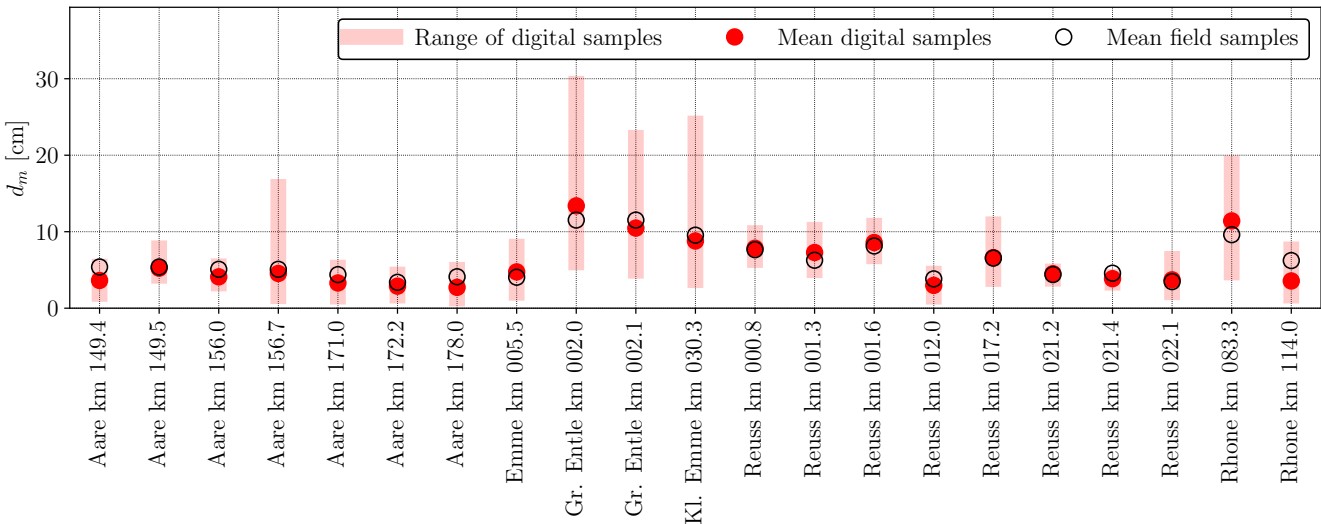

**Figure 6.** Comparison of digital line samples with 22 in-situ line samples collected in the field.

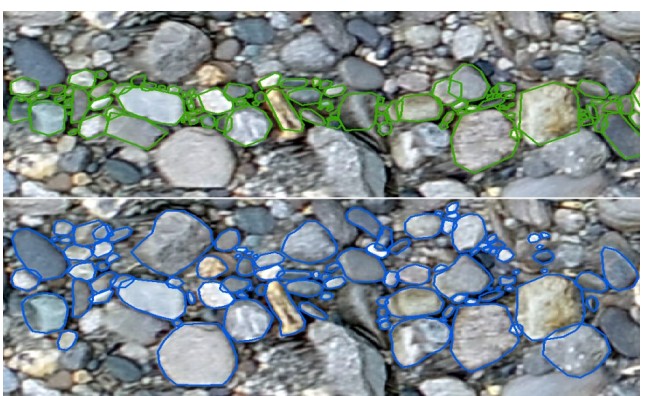

**Figure 7.** Repeated annotations of the same tile by two experts.

the best performance under a given error metric $D$ from the model trained to optimize that same metric, i.e. the best performance per column should be observed on the diagonals in the two Tables. Note that each error measure lives in its own space, numbers are not comparable across columns.

### 5.2.1 Regressing the relative frequency distribution

When estimating the relative *frequency* distribution, all three loss functions yield rather similar mean performance, in all three error metrics (Table 1 (a)). The lowest KLD (mean of 0.13) and the highest IoU (mean of 0.73) are achieved by optimizing the respective loss function, whereas the lowest EMD is also achieved by optimizing the IoU. However, all variations are within one standard deviation. The KLD is slightly more sensitive than the other loss functions, with the largest relative difference (0.13 vs. 0.16) corresponding to a 23 % increase. All standard deviations are one order of

magnitude smaller than the mean, meaning that the reported performance is not significantly affected by the specific splits into training and test sets.

### 5.2.2 Regressing the relative volume distribution

The regression performance for the relative *volume* distribution is presented in Table 1 (b). Here, the best mean performance is indeed always achieved by optimizing the respective loss function. The relative performance gap under the KLD error metric increases to 250 %, with 0.32 when trained with the KLD loss vs. 0.80 with the IoU loss. Also the standard deviation of the KLD between cross-validation folds exhibits a marked increase.

### 5.2.3 Performance depending on the *GRAINet* regression target

In comparison to the values reported in Table 1 (a), the KLD on the *volume* seems to be even more sensitive regarding the choice of the optimized loss function. Furthermore, all error metrics are worse when estimating the *volume* instead of the *frequency* distribution: the best EMD increases from 0.42 to 0.65, the KLD from 0.13 to 0.32, and the best IoU decreases from 0.73 to 0.61.

Looking at the difference between the *frequency* and the *volume* distribution, we see a general shift of the probability mass to right-hand side of the distributions, which is clearly visible in Fig. 8 c, f. While the *frequency* is generally smoothly decreasing to zero probability mass towards the larger grain size fractions of the distribution, the *volume* has a very sharp jump at the last bin (Fig. 8 c, f), where the largest grain — often only a single one (Fig 9 b, f) — has been measured.

|  | | EMD (↓) | KLD (↓) | IoU (↑) |
|---|---|---|---|---|
| | EMD | 0.43 (0.03) | 0.15 (0.02) | 0.72 (0.01) |
| Loss | KLD | 0.44 (0.05) | 0.13 (0.01) | 0.72 (0.01) |
| | IoU | 0.42 (0.04) | 0.16 (0.03) | 0.73 (0.01) |

(a)

|  | | EMD (↓) | KLD (↓) | IoU (↑) |
|---|---|---|---|---|
| | EMD | 0.65 (0.03) | 0.79 (0.24) | 0.61 (0.01) |
| Loss | KLD | 0.68 (0.05) | 0.32 (0.02) | 0.60 (0.01) |
| | IoU | 0.69 (0.05) | 0.80 (0.19) | 0.61 (0.01) |

(b)

**Table 1.** Results for *GRAINet* regressing (a) the relative *frequency* distribution and (b) the relative *volume* distribution. Mean and standard deviation (in parenthesis) for the random 10-fold cross validation. The rows correspond to the CNN models trained with the respective loss function. Arrows indicate if the error metric is minimized (↓, lower is better) or maximized (↑, higher is better).

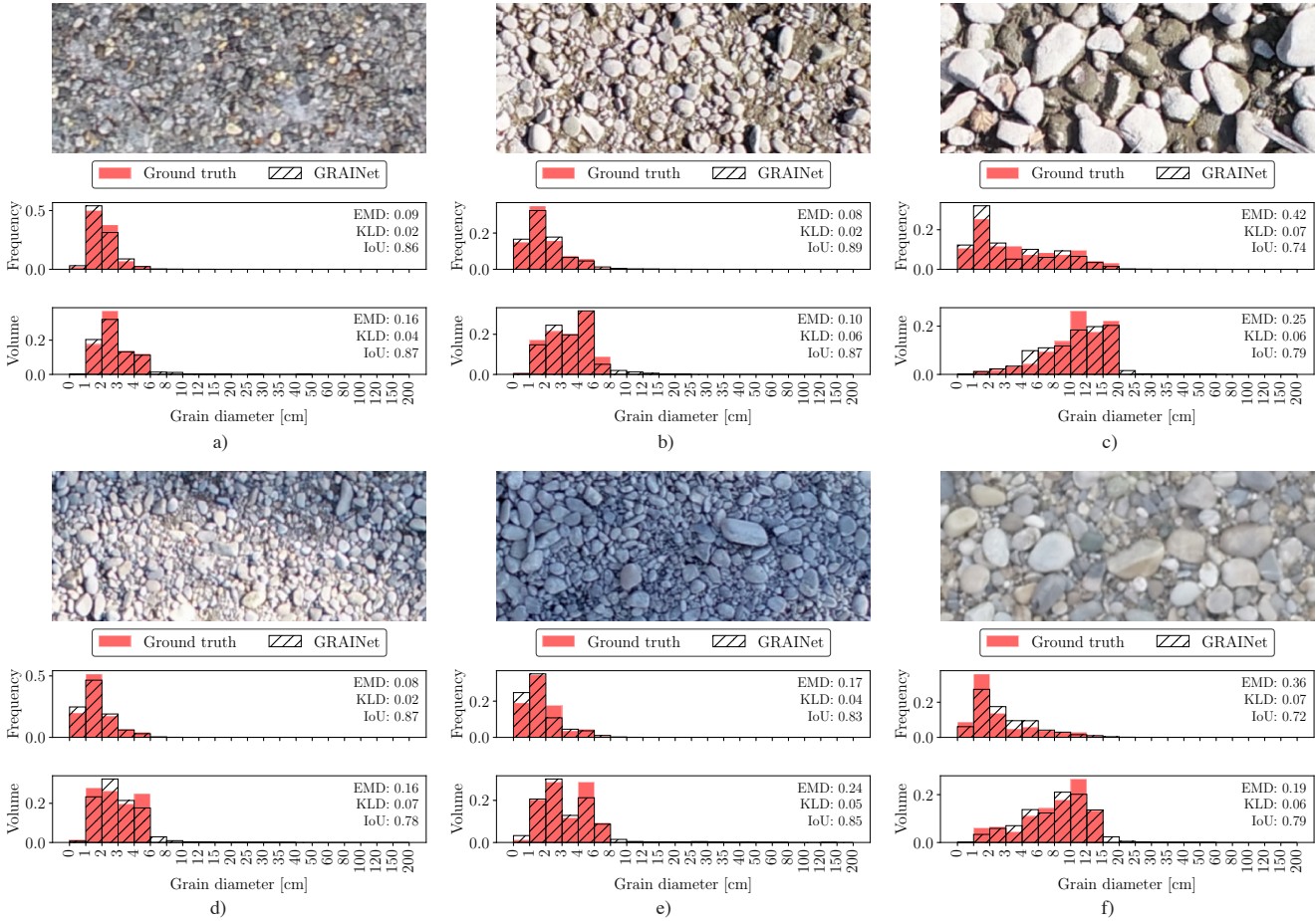

**Figure 8.** Example image tiles where the *GRAINet* regression of the relative *frequency* and *volume* distribution yields good performance.

Figure 8 displays examples of various lighting conditions and grain size distributions, where *GRAINet* yields a good performance for both targets. On the other hand, the error cases in Fig. 9 represent the limitations of the model. While e.g. extreme lighting conditions deteriorate the performance on both targets similarly (Fig 9 a), the rare radiometry caused

by moss (Fig 9 d) has a stronger effect on the *volume* prediction.

Comparing the predictions with the ground truth distributions in Fig. 9, the *GRAINet* predictions seem to be generally smoother for both *frequency* and *volume*. More specifically, the predicted distributions have longer tails (Fig 9a, c, e) and closed gaps of empty bins (Fig. 9f).

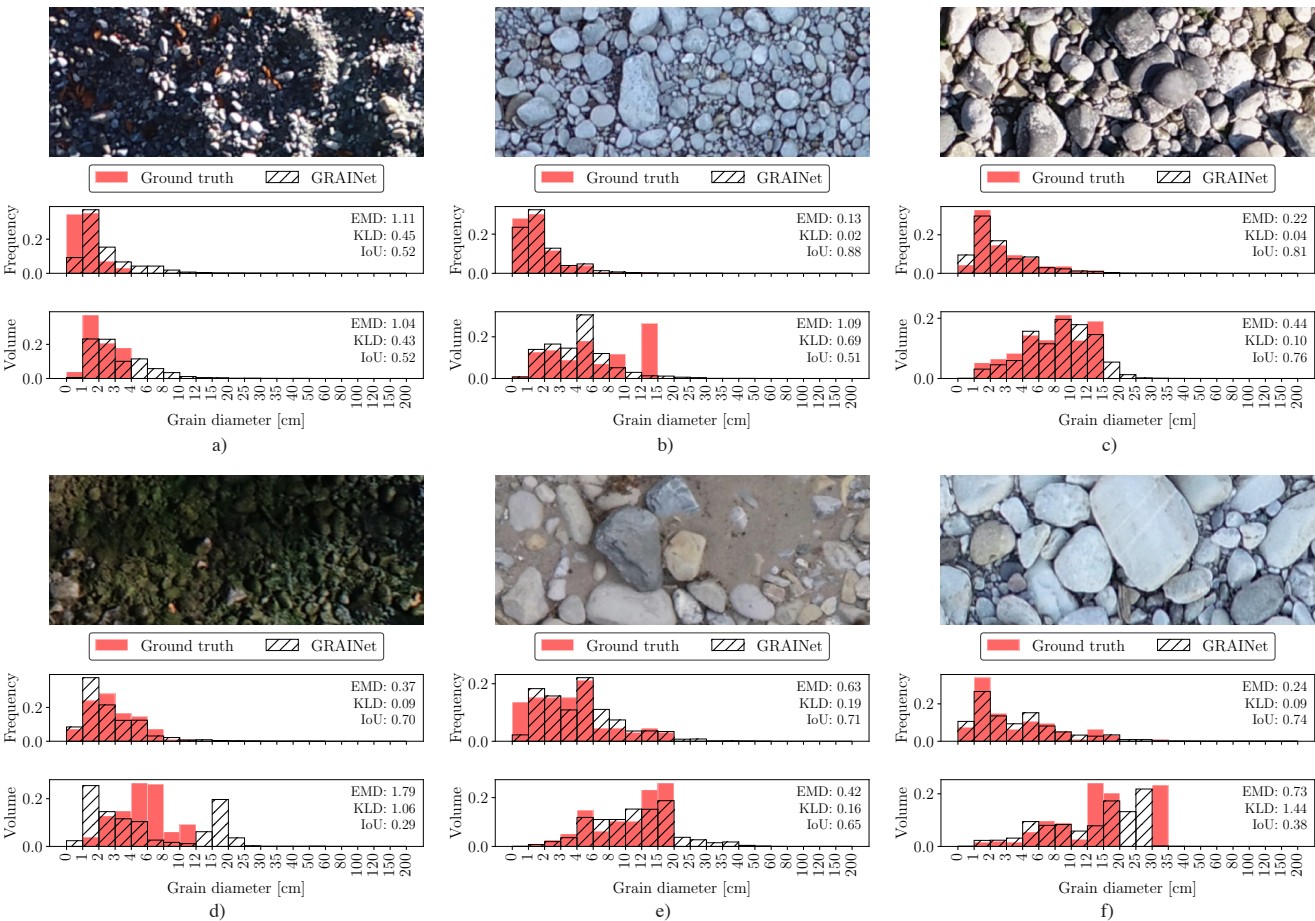

**Figure 9.** Error cases where the *GRAINet* regression of the relative *frequency* and *volume* distribution fails.

In combination with the smoother output of the CNN, the sharp jump in the *volume* distribution could be an explanation for the generally worse approximation of the *volume* compared to the *frequency*.

### 5.2.4 Learned global texture features

To investigate to what degree the texture features learned by the CNN are interpretable w.r.t. grain sizes, we visualize the activation maps of the last convolution layer, before the global average pooling, in Fig. 10. In that layer of the CNN there is one activation map per grain size bin, and those maps serve as a basis for regressing the relative frequencies. Therefore, each of these 21 activation maps corresponds to a specific bin of grain sizes, with bin 0 for the smallest grains and bin 20 for the largest ones. Light colours denote low activation, darker red denotes higher activation. To harmonize the activations to a common scale [0, 1] for visualisation, we pass the maps through a softmax over bins. This can be interpreted as a probability distribution over grain size bins at each pixel of the downsampled patch. The resulting activa-

tion maps in Fig. 10 exhibit plausible patterns, with smaller grains activating the corresponding, lower bin numbers.

### 5.2.5 Grading curves for entire gravel bars

We compute grading curves from the predicted relative *frequency* and *volume* distributions as described in Sect. 3.2. Furthermore, we average the individual curves to obtain a single grading curve per gravel bar. We show example grading curves obtained with the three different loss functions in Fig. 11. The top row shows a distribution of rather fine grains while the bottom row represents a gravel bar of coarse grains. Regarding the fine gravel bar (Fig. 11, top), the difference between the three loss functions is hard to observe. Yet, there is a tendency of overestimating the coarse fraction if optimizing for KLD. However, only KLD can reproduce the grading curve of the coarse gravel bar reasonably well (Fig. 11, bottom). Overall, the experiments indicate that the KLD loss yields best performance for all three error metrics. Thus, to assess the effect of the target choice (*frequency* vs. *volume*) on the final grading curves of all 25 gravel bars, we use the *GRAINet* trained with the KLD loss (Fig. 12).

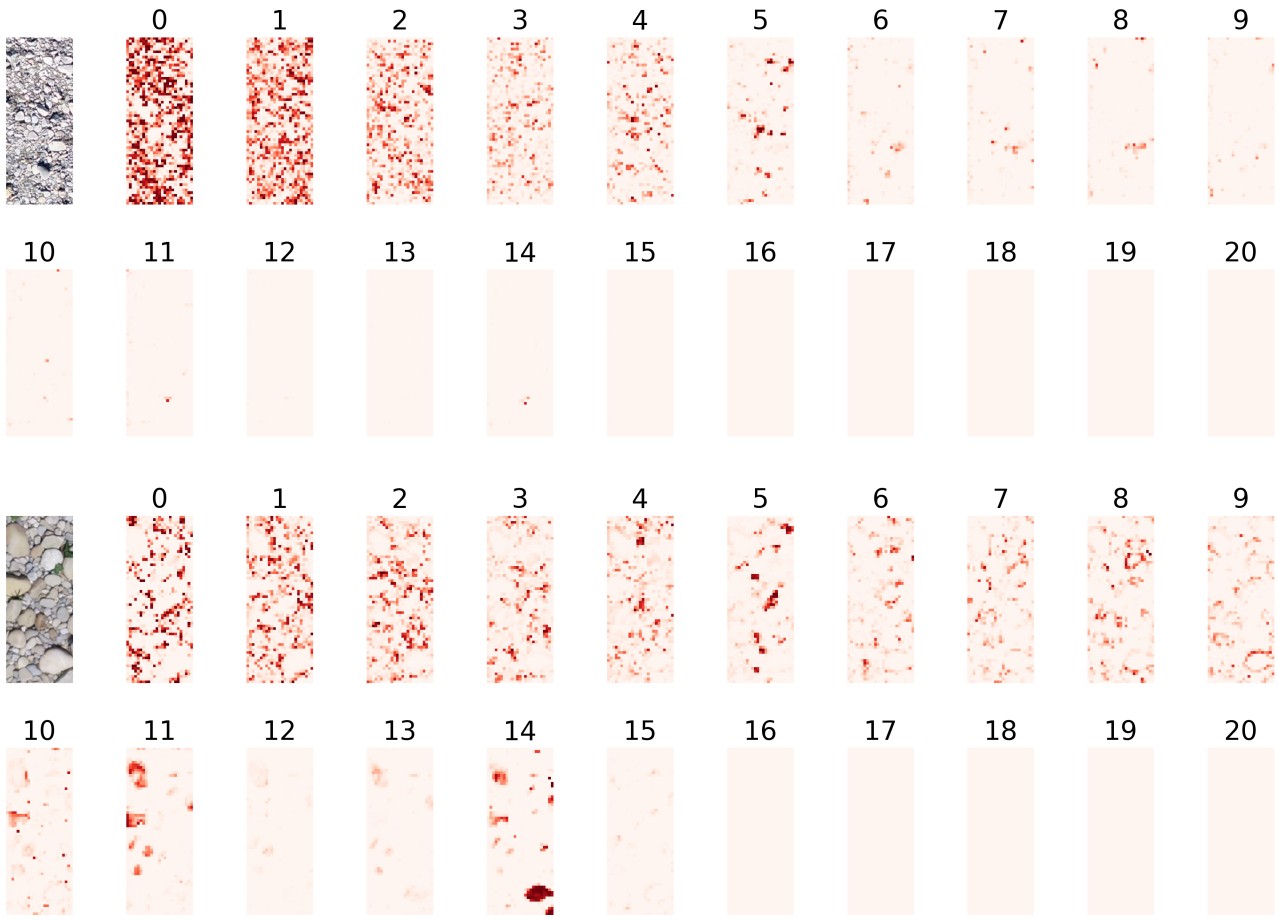

**Figure 10.** Activation maps after the last convolutional layer for two examples. Each of 21 maps corresponds to a specific histogram bin of the grain size distribution, where bin 0 corresponds to the smallest, bin 20 to the largest grains. Light colours are low activation, darker red denotes higher activation.

Both models approximate the ground truth curves well and are able to reproduce various shapes (e.g., *Aare km 171.2* vs. *Reuss km 001.3*) . However, the grading curves derived from the predicted *frequency* distribution (dashed curves) tend to overestimate higher percentiles (e.g. *Aare km 171.0*).

This qualitative comparison indicates that regressing the *volume* distribution with *GRAINet* yields slightly better grading curves than for the *frequency* distribution. If computing the grading curve from the predicted frequency distribution, small errors in the bins with larger grains are propagated and amplified in a non-linear way due to the fraction weighted transformation described in Sect. 3.2. In contrast, the *volume* distribution already includes this non-linear transformation and consequently errors are smaller.

## 5.3   Estimation of characteristic grain sizes

Characteristic grain sizes can either be derived from the predicted distributions or *GRAINet* can be trained to directly predict variables like the mean diameter $d_m$ as scalar outputs.

### 5.3.1   Regressing the mean diameter $d_m$

We again analyze the effect of different loss functions, namely the mean squared error (MSE) and the mean absolute error (MAE) when training *GRAINet* to estimate $d_m$ end-to-end, see Table 2. Note that minimising MSE is equivalent to minimising the root mean square error (RMSE). Optimizing for MAE achieves slightly lower errors under both metrics ($3.04\,\text{cm}^2$, respectively $0.99\,\text{cm}$). However, optimizing for MAE results in significantly stronger bias, with a ME of $-0.11\,\text{cm}$ (underestimation), compared to $0.02\,\text{cm}$ for the MSE. As for practical applications a low bias is considered more important, we use *GRAINet* trained with the MSE loss for further comparisons. This yields a MAE of $1.1\,\text{cm}$ ($18\,\%$), respectively an RMSE of $1.7\,\text{cm}$ ($27\,\%$). Analog to Buscombe (2013), the corresponding normalized errors in

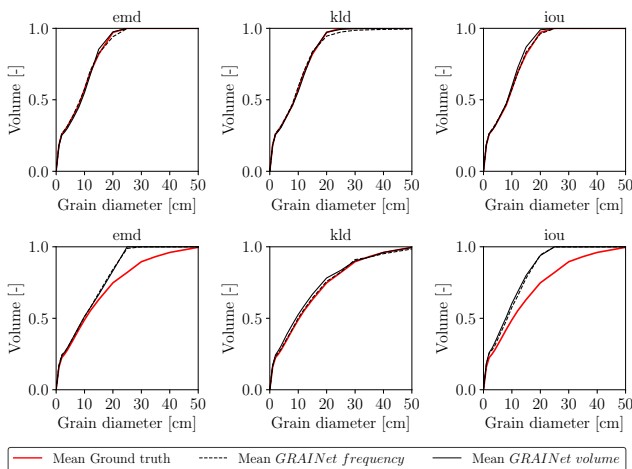

**Figure 11.** Grading curves resulting from optimizing different loss functions (from left to right: EMD, KLD, IoU) for two example gravel bars, namely *Gr. Entle km 002.0* (top) and *Reuss km 001.6* (bottom).

|      |     | MSE ($\downarrow$) | MAE ($\downarrow$) | ME (0) |
|------|-----|--------|--------|--------|
| Loss | MSE | 3.05   | 1.05   | 0.02   |
|      |     | (1.03) | (0.13) | (0.18) |
|      | MAE | 3.04   | 0.99   | $-0.11$ |
|      |     | (1.03) | (0.12) | (0.25) |

**Table 2.** Results for *GRAINet* regressing the mean diameter $d_m$ [cm]. Mean and standard deviation (in parenthesis) for the random 10-fold cross validation. The rows correspond to the CNN models trained with the respective loss function. While both MSE and MAE are minimized ($\downarrow$), the ME is optimal with zero bias (0).

parenthesis are computed by dividing through the overall mean $d_m$ of 6.2 cm (Fig. 5).

### 5.3.2   Performance for different regression targets

If our target quantity is the $d_m$, we now have different strategies. The classical multi-step approach would be to measure frequencies, convert them to volumes, and derive the $d_m$ from those. Instead of estimating the frequency, we could also directly estimate volumes, or predict the $d_m$ directly from the image data. Which approach works best? Based on the results shown so far (Table 1 and 2), we compare the $d_m$ derived from *frequency* and *volume* distributions (trained with KLD) to the end-to-end prediction of $d_m$ (trained with MSE), see Fig. 13. Regardless of the *GRAINet* target, the ME lies within +/- 0.7 cm, the MAE is smaller than 1.5 cm, and the absolute dispersion increases with increasing $d_m$. With ground truth $d_m$ values ranging from 1.3 cm to 29.3 cm, only the end-to-end $d_m$ prediction covers the full range down to 1.3 cm and up to 24 cm. In contrast, the smallest $d_m$ de-

rived from the predicted *frequency* and *volume* distribution are 2.9 cm and 2.3 cm, respectively. I.e., $d_m$-values $<3.0$ cm tend to be overestimated when derived from intermediate histograms. This is mainly due to unfavourable error propagation, as slight overestimates of the larger fractions are amplified into more serious overestimates of the characteristic mean diameter $d_m$. While the $d_m$ derived from the *volume* prediction yields a comparable MAE of 0.7 cm for ground truth $d_m$ $<3$ cm, only the end-to-end regression is able to predict extreme small, but apparently rare, values (Fig. 14). The end-to-end $d_m$ regression yields a MAE of 0.9 cm for $d_m$-values between 3 and 10 cm and 2.2 cm for values $>10$ cm.

We conclude that end-to-end regression of $d_m$ performs best. It achieves the lowest overall MAE ($<1.1$ cm) and at the same time it is able to correctly recover $d_m$ below 3.0 cm.

### 5.3.3   Mean $d_m$ for entire gravel bars

Robust estimates of characteristic grain sizes (e.g., $d_m$, $d_{50}$, etc.) for entire gravel bars or a cross-sections are important to support large scale analysis of grain size characteristics along gravel-bed rivers (Rice and Church, 1998; Surian, 2002; Carbonneau et al., 2005). To assess the performance of *GRAINet* for this purpose, the *GRAINet* end-to-end $d_m$ predictions are averaged over each gravel bar and compared with the respective mean $d_m$ of the *digital line samples* (Fig. 15). The performance averaged over all 25 gravel bars results in a MAE of 0.3 cm and a ME of 0.1 cm. The error is $<1$ cm for all gravel bars, even for the bars at the River *Gr. Entle* and *Rhone*, which have a mean ground truth $d_m$ $>10$ cm (Table C1). For 13 gravel bars the error is below $\pm0.2$ cm.

### 5.3.4   Comparison to human performance

The average standard deviation $\sigma$ of $d_m$ from repeated digital line samples accounts for 0.5 cm (see Sect. 5.1) for 17 randomly selected tiles. In comparison, regressing $d_m$ with *GRAINet* yields a root mean square error (RMSE) of 1.7 cm, of which $\approx30$ % can be explained by the label noise in the test data. We illustrate the performance of *GRAINet* versus human performance in Fig. 16. The predicted $d_m$-values lie within $1\sigma$ for 9 tiles (53 %), and within $2\sigma$ for 12 tiles (70 %).

### 5.3.5   High-resolution grain size maps

*GRAINet* offers the possibility to predict and map characteristic grain sizes densely for entire gravel bars with high resolution (1.25 m$\times$0.5 m). Three example maps are presented in Fig. 17. The mean ground truth $d_m$ per gravel bar varies between 3.0 cm (*Reuss km 012.0*, top), 3.3 cm (*Aare km 171.0*, center), and $>10$ cm (*Gr. Entle km 002.1*, bottom). For all three examples the river flows northwards.

Obviously, the map created with *GRAINet* offers full coverage of the entire gravel bar, whereas *digital line samples* deliver only a sparse map. Not only do we see that *GRAINet* successfully predicts the spatial distribution of the $d_m$ in the

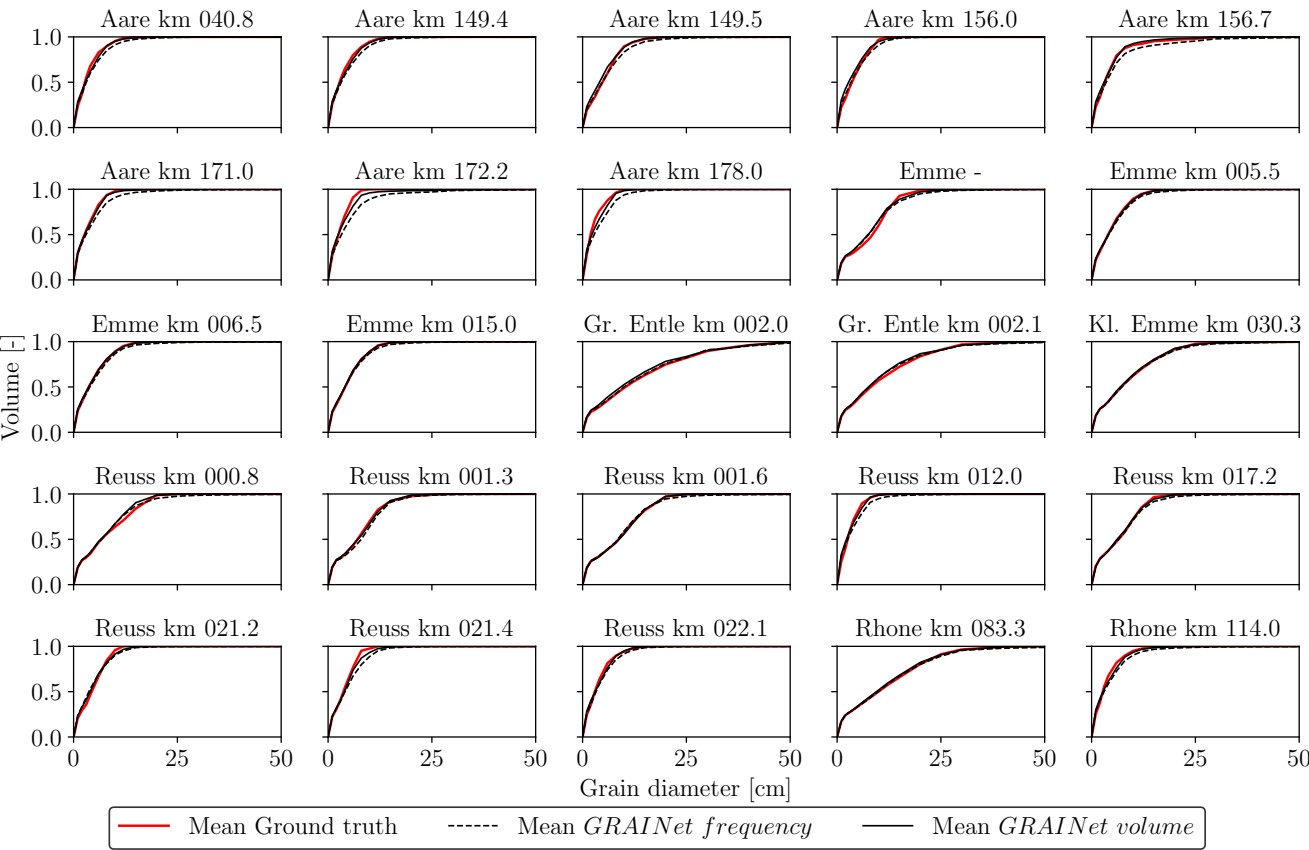

**Figure 12.** Grading curves of the 25 gravel bars, estimated with random 10-fold cross-validation.

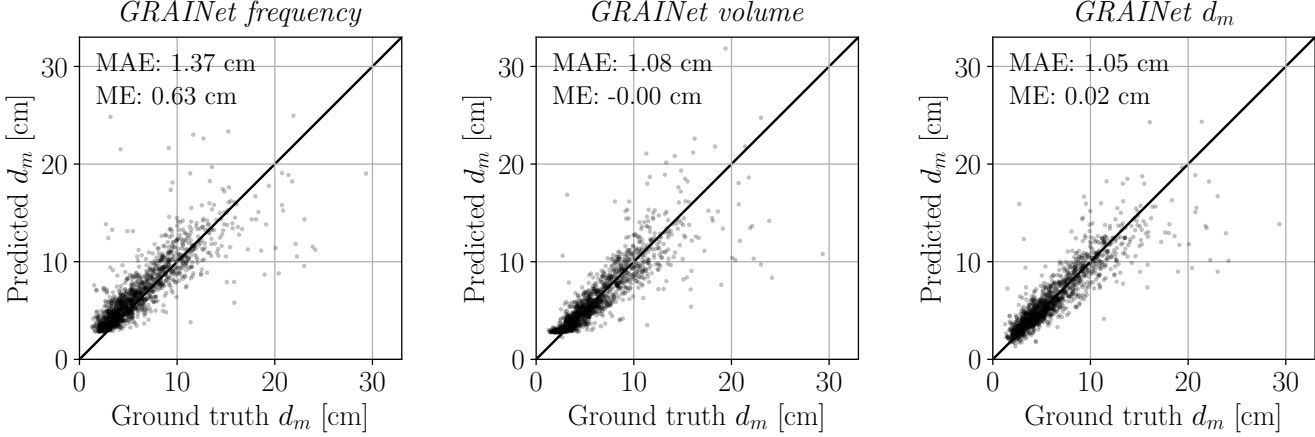

**Figure 13.** Scatter plots of the estimated $d_m$ in cm from the three *GRAINet* outputs: *frequency*, *volume*, and $d_m$ (left to right). The ground truth $d_m$ on the horizontal axis and predicted $d_m$ on the vertical axis.

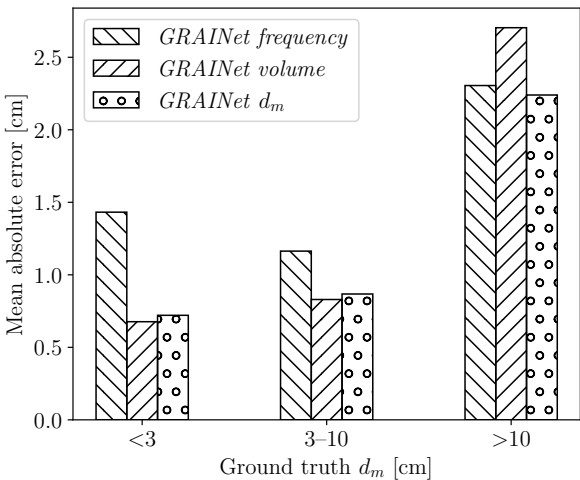

**Figure 14.** Mean absolute error of the predicted $d_m$ for three $d_m$-categories: <3 cm, 3–10 cm, and >10 cm

ground truth, it also reveals spatial patterns at a finer resolution.

Hence, *GRAINet* enables not only the assessment of difference between gravel bars but also the spatial variability and heterogeneity of $d_m$-values within a single gravel bar. Despite a similar mean $d_m$ of approximately 3 cm, the spatial layout differs greatly between *Reuss* (top) and *Aare* (center), which becomes clear when looking at dense maps of the complete gravel bars. Such sorting effects are not observable in the third example of *Gr. Entle*.

### 5.4 Generalization across gravel bars

We study the generalization capability of *GRAINet* to an unseen gravel bar with geographical cross-validation for regressing the grain size distribution and $d_m$. Note that we cannot completely isolate the effect of unseen grain size distributions from the influence of unseen imaging conditions, as each gravel bar was captured in a separate survey.

#### 5.4.1 Grading curves

Grading curves for all 25 gravel bars are given in Fig. 18. A qualitative comparison to Fig. 12 shows the effect of not seeing a single sample of the respective gravel bar during the training. The grading curves derived from the predicted *frequency* distribution seem to be less robust and overestimation of higher percentiles is increased for more than 50 % of all gravel bars. Exceptions are *Gr. Entle km 002.0* and *Rhone km 083.3*, where all percentiles are underestimated. As no striking differences are visible for about 20 gravel bars, we can say that the grading curves derived from the predicted *volume* distribution generalize (still) well in 80 % of the cases.

#### 5.4.2 Mean diameter $d_m$

We also study the generalization regarding the estimation of the $d_m$ (Fig. 19 and Fig. E1). The MAE of the random splits is <1 cm for 18 bars and <2 cm for 24 bars. When *GRAINet* is tested on unseen gravel bars (geographical cross-validation), the MAE does generally increase leading to only 15 bars <1 cm and 19 <2 cm. We observe the largest performance drop for *Aare km 156.7*, where the MAE increases from 1.4 cm (random 10-fold cross-validation) to 6 cm (geographical cross-validation). On this particular gravel bar, several tiles contain some wet and even flooded grains. Although the refraction of the shallow water could in principle change the apparent grain size, it is most likely *not* the main reason for the poor generalization. Rather, the model has simply not learned the radiometric characteristics of wet grains, as there aren't any among the samples from the other bars, used for training.

### 5.5 Effect of the image resolution

*GRAINet* does not explicitly detect individual grains, but learns to identify global texture patterns. It thus seems feasible to apply *GRAINet* to images of lower resolution, where individual grains would no longer be recognizable by a human annotator (see example in Fig. 20). To simulate that situation, we bilinearly downsample the original image resolution of 0.25 cm by factors 2, 4, 8, 16, 32, and 40, corresponding to pixel sizes of 0.5, 1.0, 2.0, 4.0, 8.0, and 10.0 cm, respectively. The CNN model is then trained and evaluated at each resolution separately. When regressing *frequency* or *volume* distribution, the performance decreases rather continuously with decreasing resolution (Fig. F1 in the appendix). Interestingly, the performance for regressing $d_m$ with *GRAINet* drops only after downsampling with factor 16 (4 cm resolution) to a MAE of 1.4 cm and reaches a MAE of 1.9 cm at factor 40 (10 cm resolution) (Fig. 21). Corresponding $d_m$ scatter plots are shown in Fig. F2, where dispersion grows with coarser resolution. Avoiding the explicit detection of individual grains with the proposed regression approach has great potential and allows us to make reasonable predictions of $d_m$ even at lower image resolutions, and to adapt the resolution to the accuracy requirements of the application. In contrast to Carbonneau (2005), our *GRAINet* is able to predict mean diameters smaller than the ground sampling distance (Fig. F2), taking a big step towards grain size mapping beyond the image resolution. We believe that, in principle, *GRAINet* could even be used to process airborne imagery from country-wide flight campaigns, depending on the accuracy requirements of the application.

### 6 Discussion

We have shown that *GRAINet* is able to estimate the full grain size distribution at particular locations in the orthophoto.

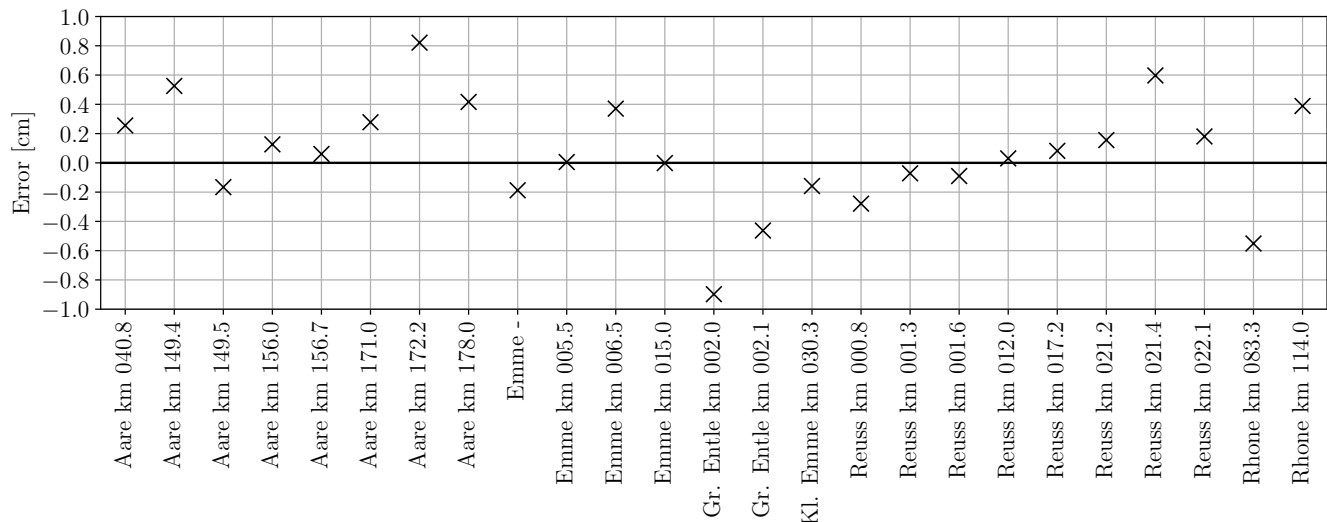

**Figure 15.** Error of the mean $d_m$ per gravel bar derived from the *GRAINet* end-to-end $d_m$ predictions.

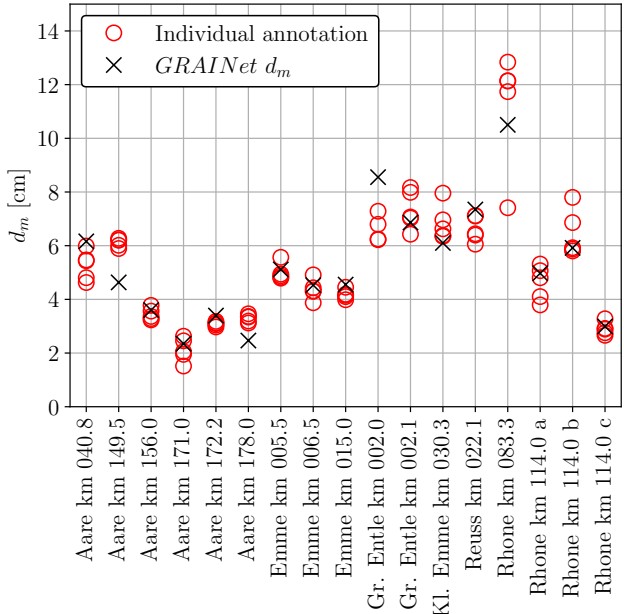

**Figure 16.** Variation of $d_m$ annotated by three to five different human experts, compared to end-to-end $d_m$ regression of *GRAINet*.

Hence, we can derive the mean grading curve of entire gravel bars. The same architecture can also be trained to densely map the spatial distribution of the $d_m$.

### 6.1 Manual component of the presented approach

5 Obviously, creating a large, manually labeled training dataset is time-consuming, a property our CNN shares with other supervised machine learning methods. However, at test time

the proposed approach requires no parameter tuning by the user, a considerable advantage for large-scale applications, where traditional image processing pipelines struggle, since 10 they are fairly sensitive to varying imaging conditions. Semi-automatic image labeling with the support of traditional image processing tools (Detert and Weitbrecht, 2012; Purinton and Bookhagen, 2019) might be an alternative way to speed up this annotation process. However, one would have 15 to carefully avoid systematic algorithmic biases in the semi-automatic procedure, otherwise the CNN will almost certainly learn to faithfully reproduce those biases. Manual (re)labeling would still be required to prevent the CNN from replicating the systematic biases and failures of the rule- 20 based system, but could be limited to challenging samples. Similarly, systematic behaviours of specific annotators may also be learned by the model. Ideally, training data should thus be generated by different skilled annotators.

The CNN predictions for a full orthophoto are masked 25 manually to the gravel bars. Our CNN is only trained on gravel images and did not see any purely non-gravel images patches with, e.g., vegetation, sand, or water. Consequently such inputs lie far outside the training distribution and result in arbitrary predictions that need to be masked out by 30 the user. The network could also be trained to ignore samples with land cover other than gravel, but this is beyond the scope of the present paper. It could be added in the future to further reduce manual work.

### 6.2 Geographical generalization                              35

We present experiments to evaluate the generalization of our approach to new locations, i.e. unseen gravel bars. In this setup, the data is exploited best, allowing the CNN to learn features invariant to the imaging conditions by providing 24

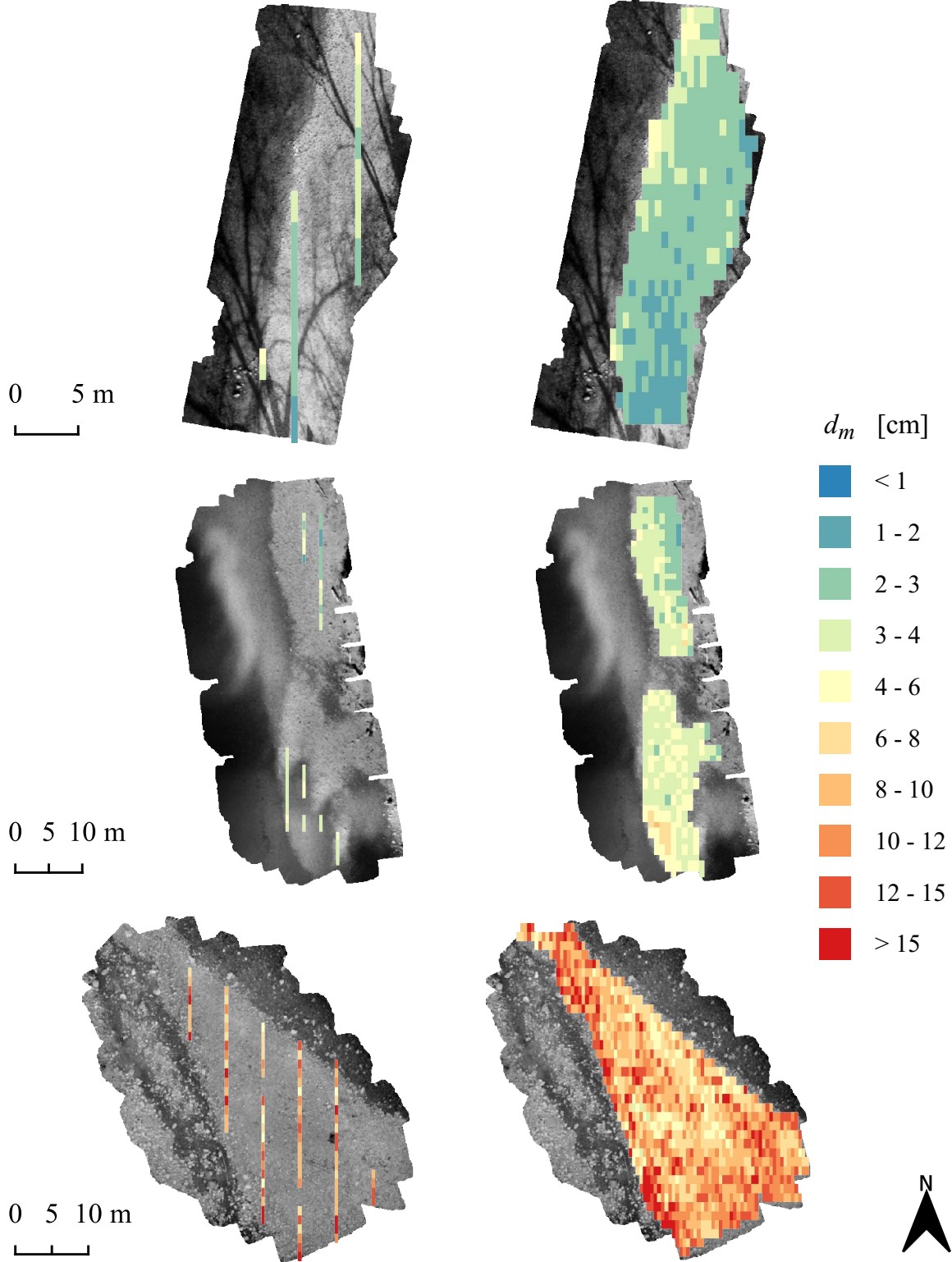

**Figure 17.** Maps of the characteristic mean diameter $d_m$, Hand-labeled ground truth, i.e. *digital line samples* (left) and *GRAINet* end-to-end $d_m$ predictions (right). Gravel bars (top to bottom): *Reuss km 012.0*, *Aare km 171.0*, and *Grosse Entle km 002.1*. The background is a gray scale version of the input UAV image.

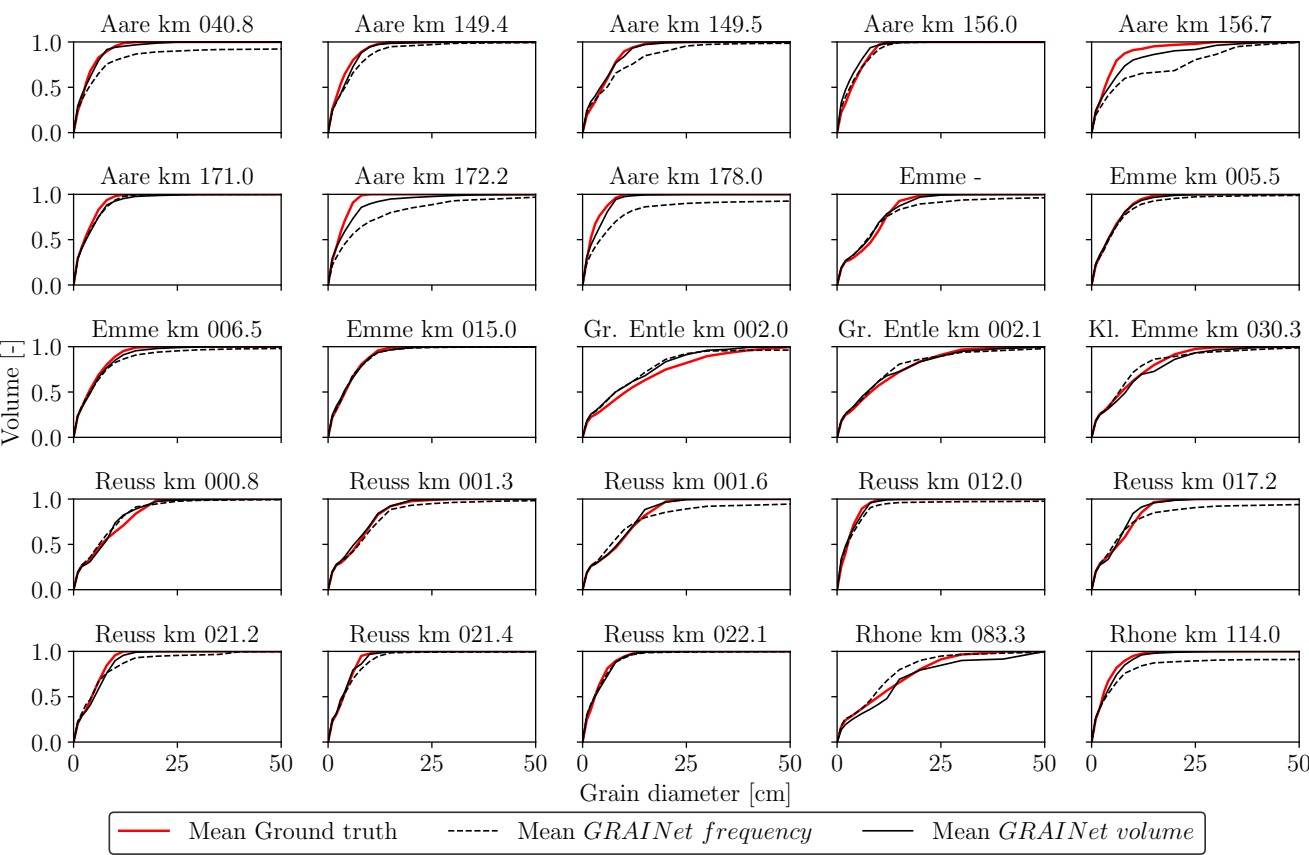

**Figure 18.** Grading curves of the 25 gravel bars, estimated with geographical cross-validation.

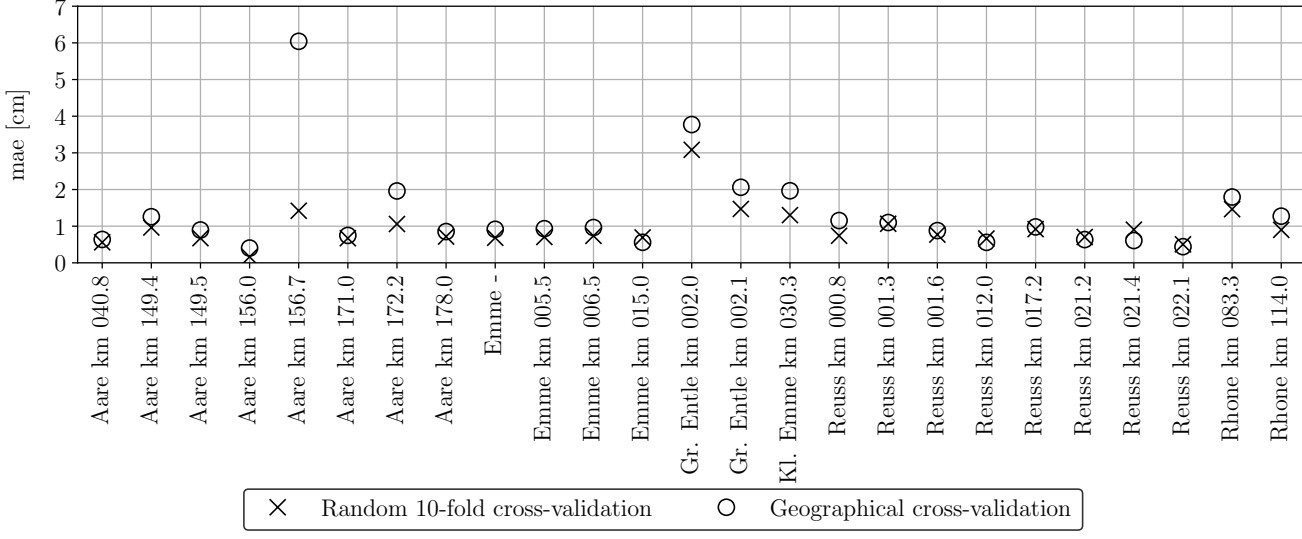

**Figure 19.** Mean absolute error of the *GRAINet* end-to-end $d_m$ regression per gravel bar (random – and geographical cross-validation).

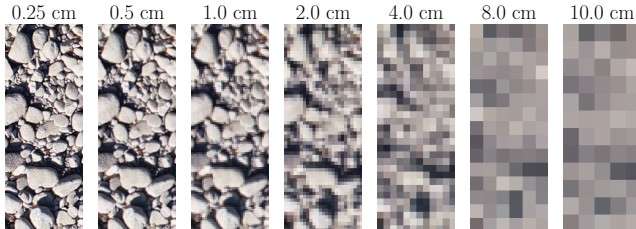

0.25 cm    0.5 cm    1.0 cm    2.0 cm    4.0 cm    8.0 cm    10.0 cm

**Figure 20.** Example image tile resampled to lower resolutions. Left to right: full resolution of 0.25 cm and tested downsampling factors: 2, 4, 8, 16, 32, 40 corresponding to: 0.5, 1.0, 2.0, 4.0, 8.0, 10.0 cm.

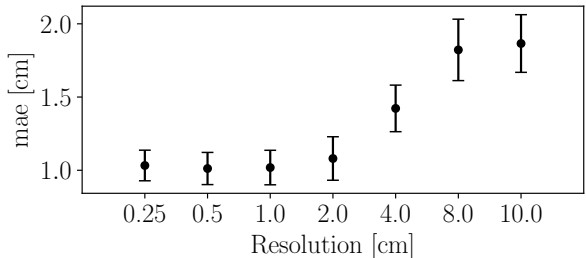

**Figure 21.** Performance of *GRAINet* $d_m$ regression with different image resolutions (trained with MSE loss).

different training orthophotos in each experiment. That experimental setup is valid to investigate geographical generalization, since there is no strong correlation between bars from the same river. An alternative experiment would be to hold-out all bars from a specific river for testing. This might be necessary in some geographical conditions with slowly varying river properties to avoid any misinterpretation and overly optimistic results. We have compared the average performance drop in the generalization experiment between five gravel bars on individual river reaches, i.e., bars that are separated by tributaries with new input of sediment, against all bars. Both groups yield comparable performance drops. We conclude, within our dataset, seeing bars from the same river during training does not lead to over-optimistic results (see Fig. H1). Generalization is mainly affected by unique local environmental factors (e.g. wet stones, algae covering) that were not seen during training. Thus, in our case, we favour the former to maximize the number of training samples as well as the number of drone surveys with varying imaging conditions and local environmental factors.

The per bar generalization experiment is furthermore justified by the fact that the characteristics of the investigated gravel bars vary greatly along the same river, both quantitatively (mean $d_m$, $d_m$ range in Table C1) and qualitatively (see appendix Fig. G1, where tiles are grouped by river name). Not only is this due to the distance between the bars, but also due to the changing slope and the varying river bed widths in mountain environments (Reuss, Aare, Emme). Furthermore, the bars are geographically separated through trib-

utaries (Aare, Rhone), leading to a drastic increase in the catchment areas between the bars. E.g., at the river Rhone the catchment area is more than doubled from $982\,km^2$ (at km 083.3) to $2485\,km^2$ (at km 114.0). Additionally, the sediment transport is affected by dams at the rivers Aare, Rhone, and Reuss. Finally, the characteristics may also be artificially altered, as it is nowadays common in Central Europe to replenish gravels of 2-3 cm to create spawning grounds for fish. For instance, the grain size distribution at the bar Reuss km 022.1 (and probably km 012.0) is very likely affected by such a targeted replenishment of sediment.

If we were to hold out, say, the whole river Aare, we would not only substantially reduce the number of training samples, but also the diversity of imaging conditions. In fact, within our experimental setup we already present one hold-one-river-out experiment for the river Kl. Emme, from which only one bar is included in our dataset. Even though this bar contains the largest number of digital line samples, its estimated grain size distribution fits rather well in the geographical cross-validation experiment (see Fig. 18). The observed performance drop between the random and geographical cross-validation experiment for individual bars in Fig. 19 is rather explained by coarse gravel bars with a large mean $d_m$ and a wide $d_m$ range. Seeing bars from the same river during training and testing does not seem to have an effect (for instance, Aare).

Ultimately, it is important to keep in mind that data-driven approaches, like the one proposed, will only give reasonable estimates if the test data approximately matches the training data distribution. It will not perform well for out-of-distribution (OOD) samples. Detecting such OOD samples is an open problem and an active research direction.

## 6.3 Comparison to previous work

While, existing statistical approaches are limited to output characteristic grain sizes ($d_m$, $d_{50}$), to the best of our knowledge, GRAINet is the first data-driven approach that is able to regress a full, local grain size distribution at each location in an orthophoto. We are neither aware of any previous work that evaluates grain size estimation over entire gravel bars in river beds, nor of a comparable study regarding geographical generalization.

Nevertheless, we present a generic learning approach, i.e. the same architecture can also be trained to directly predict other desired grain size metrics derived from the distribution, such as the mean diameter $d_m$. Due to the end-to-end learning, our proposed CNN approach is able to extract global texture features that are informative about grain size beyond the image resolution, and thus beyond the sensitivity of human photo-interpretation or traditional image processing that relies on local image gradients to delineate individual grains. Even the latest work of Purinton and Bookhagen (2019) can only detect individual grains that have a $b$-axis $20\times$ the ground sampling distance. Also previous statistical

approaches based on global image texture (Carbonneau et al., 2004) are limited by the input resolution and can only predict the median diameter $d_{50}$ down to 3 cm at a comparable spatial resolution of 1 m. Hence, we believe that our approach advances the state of the art.

A direct quantitative comparison to previous work with different application focus and different data is only possible to a limited extent. E.g., Buscombe (2013) evaluate on a mixed dataset with samples from rivers, natural beach, and continental shelf sediments and report normalized mean absolute errors of the estimated percentiles ranging from 10 to 29 %. In comparison, our $d_m$ regression yields a normalized mean absolute error of 18 %.

## 6.4 Advantages and limitations of the approach

Our CNN-based approach makes it possible to robustly estimate grain size distributions and characteristic mean diameters from raw images. By analyzing global image features, *GRAINet* avoids the explicit detection of individual grains, which makes the model more efficient and leads to a robust performance, even with lower image resolutions. The proposed approach enables the automatic analysis of entire gravel bars without destructive measures and with reasonable effort.

Advantages are manifold. First, results are objective and reproducible, as they are not influenced by a subjectively chosen sampling location and grain selection. Second, the resulting curves and $d_m$ represent the whole variability of grains of a gravel bar and thus the disproportionately high effect of single coarse grains on the curve and on $d_m$ can be reduced. Consequently, the derived mean curves and its characteristic grain sizes can be considered representative.

Our experiments highlight some limitations due to the limited sample size for training. While 10-fold cross-fold validation yields very satisfying results, the poorer performance of the geographical cross-validation reveals that collecting and annotating sufficiently large and varied training sets is essential. Unseen unique local environmental factors such as wet stones or algae covering caused performance drops in the generalization experiment. However, if the model has seen a few of these samples (random cross-validation) the performance is more robust against such disturbances. Additionally, the performance of *GRAINet* deteriorates for very coarser gravel bars, as indicated by the error metrics of the distributions as well as for $d_m$. The lower performance is caused by larger variability and by the high impact of individual, large grains, as well as by the unbalanced data distribution (only 14 % of the digital line samples have a $d_m$ >10 cm). Application of *GRAINet*, trained with our dataset, is thus not always satisfactory for coarse, unseen gravel bars. In order to improve results and extend potential applications fields, further digital line samples from additional UAV surveys should be collected.

Finally, the best performance has been achieved with high-resolution imagery taken at 10 m flying altitude. At this altitude it takes approximately 15 minutes to cover an area of one hectare with a DJI PHANTOM 4 PRO (that has a max. flight time of approx. 30 min per battery). It would be advantageous to reduce the flight time per area by flying at higher altitudes. As our resolution study on artificially downsampled images shows, the CNN may yield satisfactory performance on images with 1–2 cm resolution corresponding to 40–80 m flying altitude. While this is a promising result, it remains to be tested on images taken at such flying altitudes. We expect that retraining the model with high altitude image-label pairs will lead to similar performance as in the artificial case.

## 6.5 Potential applications

The presented *GRAINet* method can be applied to all rivers that fulfil the following conditions: dry gravel bars (meaning low water conditions, as grains in deeper water cannot be analyzed) and no obstacles in the flight area (especially trees along the rivers can cause occlusions). Despite these limitations, we are convinced that our results confirm the large potential of UAV surveys in combination with CNNs for grain size analysis. There are several applications which become possible with *GRAINet* in a quality that was hitherto not achievable. Perhaps the greatest asset is the creation of dense, spatially explicit, geo-referenced maps of $d_m$. Not only can they help to understand spatial sorting effects of bedload transport processes, they can also be used to calibrate two- or even three-dimensional fractional transport models. In addition, the variability of $d_m$ within a gravel bar can provide important information regarding the bedload regime and the ecological value of the river (e.g., as aquatic habitat). For example, a lack of variability in the finer grain sizes is a clear sign for bed armouring and thus an important indicator for a bedload deficit. Consequently, the maps of $d_m$ are ideal for large-scale monitoring in space and time, since they open up the possibility to study entire bars or river branches at virtually no additional cost. Our automatic approach handles all samples consistently and allows for unbiased monitoring over long times, as there is no variation due to changing operators (Wohl et al., 1996). Furthermore, the ability to estimate mean diameters from lower image resolutions (up to 2 cm ground sampling distance) will allow to cover even larger regions flying the UAV at higher altitudes. Due to the high resolution of the resulting maps and distribution estimates, local effects on bars can be investigated. Ultimately, this could allow hydrologists to explore new research directions that advance the understanding of fluvial geomorphology. While spatially explicit data may lead to an improved calibration of numerical models, we may gain new insights into how spatial heterogeneity affects the sediment transport capacity as well as the aquatic biodiversity.

## 7 Conclusions and future work

We have presented *GRAINet*, a data-driven approach centered on deep learning to analyze grain size distributions from georeferenced UAV images with a convolutional neural network. In an experimental evaluation with 1,491 *digital line samples*, the method achieves an accuracy that makes it relevant for several practical applications. The new possibility to carry out holistic analyses of entire gravel bars overcomes the limitations of sparse field sampling approaches (e.g. *line sampling* by Fehr (1987)), which is cumbersome and prone to subjective biases.

As CNNs are generic machine learning models, they offer great flexibility to directly predict other variables, like for example the ratio $d_{84}/d_{16}$ or other specific percentiles (Buscombe, 2019). In fact, it might be promising to design a multitask approach, in order to exploit the correlations and synergies between different variables and parametrizations describing the same grain size distribution. Obviously, collecting more training data can be expected to benefit the generalization performance of *GRAINet*. Data annotation could potentially be supported with *active learning* (e.g. Settles, 2009), where the model is gradually updated and intermediate predictions guide the selection of the most informative samples that should be labeled to further improve the model. Another technically interesting direction to explore is *domain adaptation*, in order to exploit *un*labeled image data as a source of information and improve the generalization to a new domain with potentially different characteristics (i.e., new gravel bars).

*Code and data availability.* The code with a demonstration on a subset of the data is available on: github.com/langnico/GRAINet. Due to licensing restrictions, the complete dataset may only be used for research purposes and can be requested by contacting andrea.irniger@hzp.ch.

*Author contributions.* NL and AR developed the code and carried out the experiments. AI designed the data acquisition and analyzed the results. KS, JW and RH provided guidance during project planning and experimentation. All authors contributed to the manuscript, under the lead of NL and AI.

*Competing interests.* The authors declare that they have no conflict of interest.

*Acknowledgements.* We thank Hunziker, Zarn & Partner for sharing the ground truth data for this research project.

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

**Appendix B:  CNN architecture illustration**

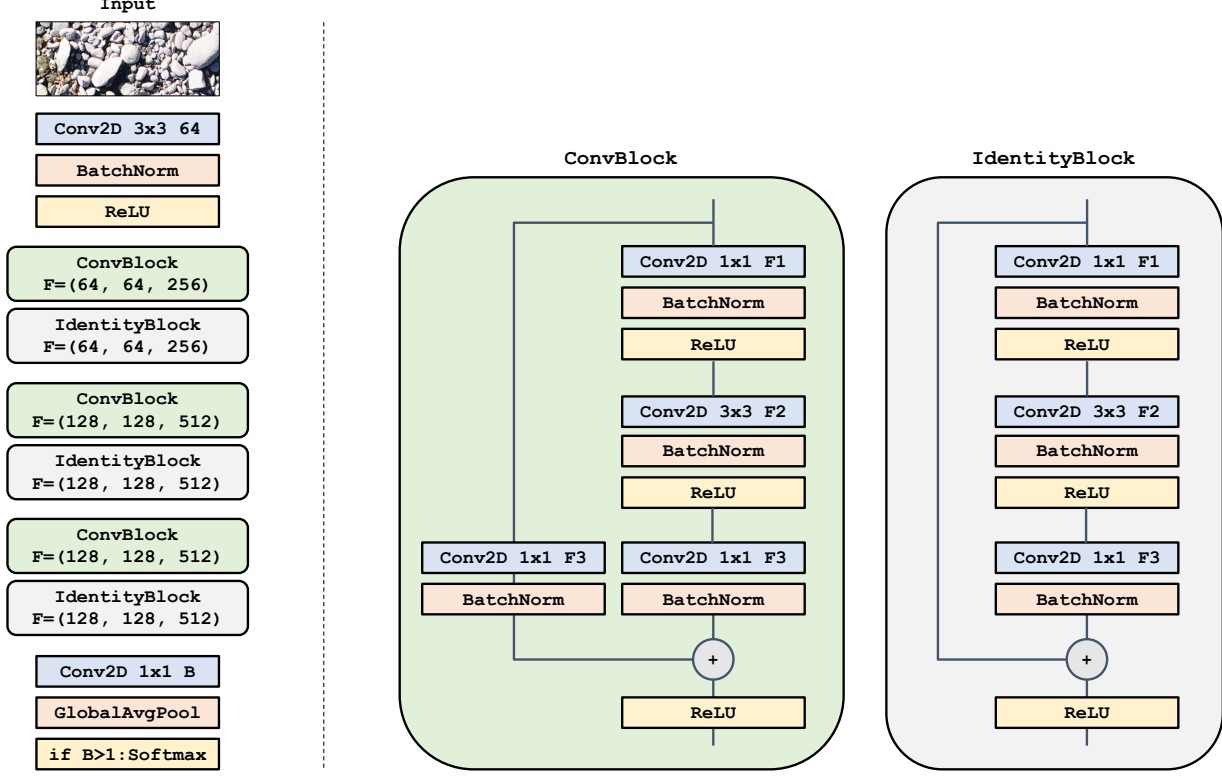

**Figure B1.** Illustration of the convolutional neural network architecture. Left: Full architecture where $F = (F1, F2, F3)$ denotes the number of filters of the three convolutional layers within the residual blocks. The number of outputs $B$ corresponds to the estimated histogram bins. To estimate a scalar (e.g. mean diameter) this B can be set to one. Right: The respective residual blocks. The *ConvBlock* has a learnable skip connection and is used when the number of output features is not equal to the number of input features. If constant, the *IdentityBlock* forwards the input.

**Appendix C:  Overview of investigated gravel bars**

| Gravel bar name | Slope [%] | Bed width [m] | Annual mean runoff [m³/s] | Num. labels [-] | Mean $d_m$ [cm] | Min $d_m$ [cm] | Max $d_m$ [cm] | Range $d_m$ [cm] |
|---|---|---|---|---|---|---|---|---|
| Aare km 040.8 | 0.02 | 75 | 221 | 114 | 3.5 | 1.7 | 8.5 | 6.8 |
| Aare km 149.4 | 0.07 | 100 | 172 | 30 | 3.6 | 1.9 | 5.4 | 3.6 |
| Aare km 149.5 | 0.07 | 100 | 172 | 12 | 5.2 | 4.2 | 7.8 | 3.6 |
| Aare km 156.0 | 0.02 | 75 | 122 | 5 | 4.1 | 3.2 | 5.5 | 2.2 |
| Aare km 156.7 | 0.02 | 75 | 122 | 40 | 4.5 | 1.6 | 15.9 | 14.3 |
| Aare km 171.0 | 0.12 | 60 | 122 | 42 | 3.3 | 1.5 | 5.3 | 3.8 |
| Aare km 172.2 | 0.12 | 60 | 122 | 60 | 2.9 | 1.7 | 4.4 | 2.8 |
| Aare km 178.0 | 0.12 | 60 | 122 | 32 | 2.7 | 1.3 | 5.0 | 3.8 |
| Emme km - | - | - | - | 14 | 7.9 | 6.3 | 9.5 | 3.2 |
| Emme km 005.5 | 0.46 | 50 | 14 | 190 | 4.7 | 2.0 | 8.0 | 6.0 |
| Emme km 006.5 | 0.46 | 50 | 14 | 116 | 4.6 | 2.5 | 7.7 | 5.2 |
| Emme km 015.0 | 0.59 | 75 | 14 | 78 | 4.8 | 2.9 | 8.9 | 6.0 |
| Grosse Entle km 002.0 | 1.50 | 60 | 2.5 | 76 | 13.4 | 6.0 | 29.3 | 23.4 |
| Grosse Entle km 002.1 | 1.50 | 60 | 2.5 | 90 | 10.5 | 4.9 | 22.3 | 17.4 |
| Kleine Emme km 030.3 | 1.40 | 63 | 6.2 | 212 | 8.8 | 3.7 | 24.1 | 20.5 |
| Reuss km 000.8 | 0.14 | 70 | 140 | 22 | 7.8 | 6.3 | 9.8 | 3.5 |
| Reuss km 001.3 | 0.14 | 70 | 140 | 16 | 7.3 | 5.0 | 10.3 | 5.3 |
| Reuss km 001.6 | 0.14 | 70 | 140 | 22 | 8.6 | 6.8 | 10.8 | 4.0 |
| Reuss km 012.0 | 0.17 | 80 | 140 | 34 | 3.0 | 1.5 | 4.5 | 3.0 |
| Reuss km 017.2 | 0.19 | 110 | 140 | 66 | 6.6 | 3.8 | 11.0 | 7.2 |
| Reuss km 021.2 | 0.18 | 60 | 140 | 6 | 4.5 | 3.9 | 4.8 | 1.0 |
| Reuss km 021.4 | 0.18 | 60 | 140 | 4 | 3.9 | 3.3 | 4.1 | 0.8 |
| Reuss km 022.1 | 0.18 | 60 | 140 | 30 | 3.7 | 2.1 | 6.4 | 4.4 |
| Rhone km 083.3 | 1.50 | 100 | 110 | 74 | 11.4 | 4.7 | 19.0 | 14.3 |
| Rhone km 114.0 | 0.18 | 60 | 42 | 106 | 3.6 | 1.7 | 7.7 | 6.0 |

**Table C1.** Overview of the 25 investigated gravel bars. From left to right: Slope and bed width of the river at this location with the corresponding annual mean water runoff. Number of annotated image tiles (*digital line samples*). Ground truth statistics of the characteristic mean diameter $d_m$.

**Appendix D: Repeated human annotation**

| Image tile name | Mean | Standard deviation | Min | Max | Range |
|---|---|---|---|---|---|
| Aare km 040.8 | 5.3 | 0.5 | 4.6 | 6.0 | 1.4 |
| Aare km 156.0 | 3.4 | 0.2 | 3.2 | 3.8 | 0.5 |
| Aare km 149.5 | 6.1 | 0.1 | 5.9 | 6.3 | 0.4 |
| Aare km 172.2 | 3.1 | 0.1 | 3.0 | 3.2 | 0.2 |
| Aare km 171.0 | 2.1 | 0.4 | 1.5 | 2.6 | 1.1 |
| Aare km 178.0 | 3.3 | 0.1 | 3.1 | 3.5 | 0.3 |
| Emme km 005.5 | 5.0 | 0.3 | 4.8 | 5.6 | 0.8 |
| Emme km 006.5 | 4.4 | 0.3 | 3.9 | 4.9 | 1.0 |
| Emme km 015.0 | 4.2 | 0.2 | 4.0 | 4.5 | 0.5 |
| Grosse Entle km 002.0 | 6.6 | 0.4 | 6.2 | 7.3 | 1.1 |
| Grosse Entle km 002.1 | 7.3 | 0.6 | 6.4 | 8.2 | 1.7 |
| Kleine Emme km 030.3 | 6.9 | 0.6 | 6.3 | 8.0 | 1.6 |
| Reuss km 022.1 | 6.6 | 0.4 | 6.1 | 7.1 | 1.1 |
| Rhone km 083.3 | 11.3 | 2.0 | 7.4 | 12.8 | 5.4 |
| Rhone km 114.0 a | 6.5 | 0.8 | 5.8 | 7.8 | 2.0 |
| Rhone km 114.0 b | 4.6 | 0.6 | 3.8 | 5.3 | 1.5 |
| Rhone km 114.0 c | 2.9 | 0.2 | 2.7 | 3.3 | 0.6 |

**Table D1.** Statistics of the mean diameter $d_m$ [cm] from three to five human annotations for 17 randomly selected image tiles.

## Appendix E: Generalization across gravel bars

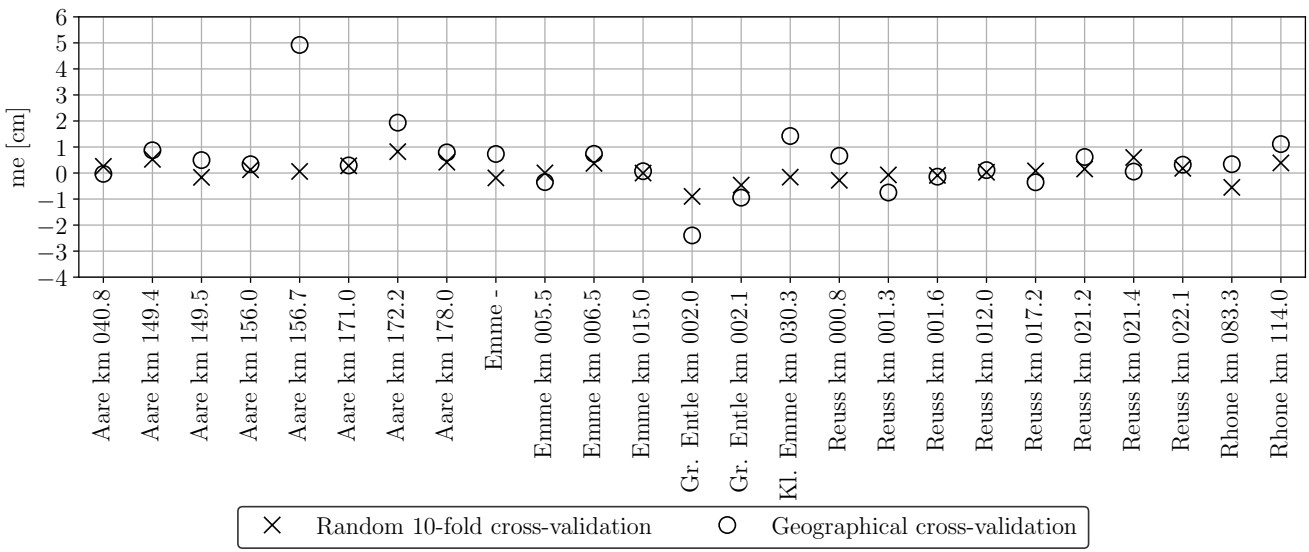

**Figure E1.** Mean error (bias) of the *GRAINet* end-to-end $d_m$ regression per gravel bar (random – and geographical cross-validation).. A positive error implies that the *GRAINet* prediction was higher than the ground truth.

**Appendix F:  Resolution study**

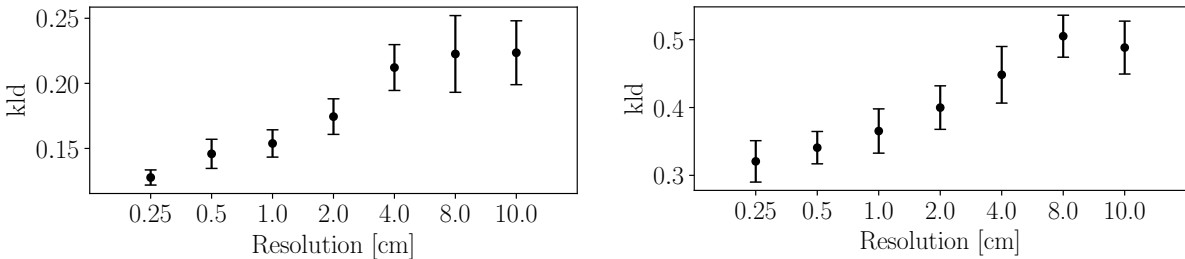

**Figure F1.** Resolution study for *GRAINet* regressing the relative *frequency* distribution (left) and regressing the relative *volume* distribution (right) by optimizing the KLD loss.

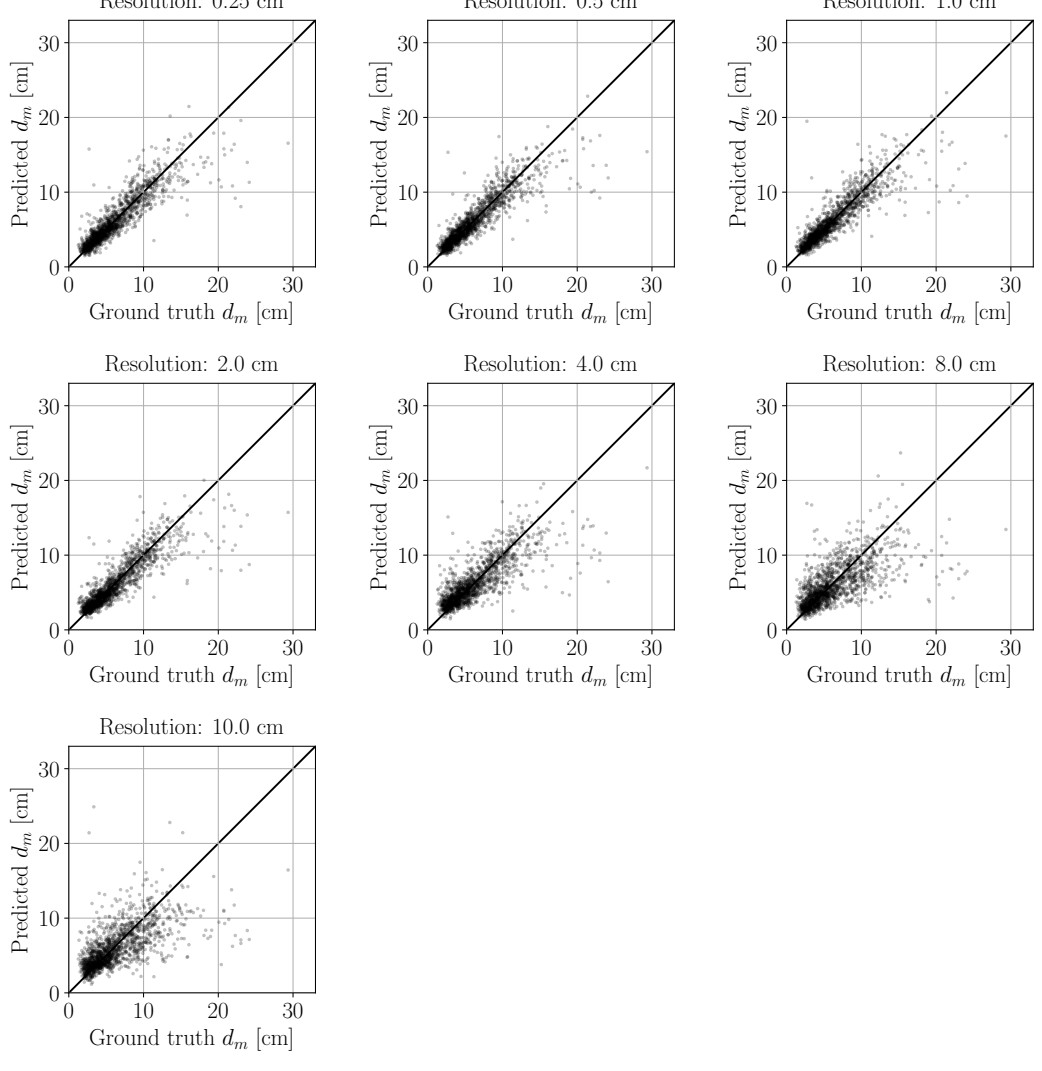

**Figure F2.** Scatter plots of *GRAINet* $d_m$ regression with different image resolutions (trained with MSE loss).

**Appendix G: Tiles grouped by river name**

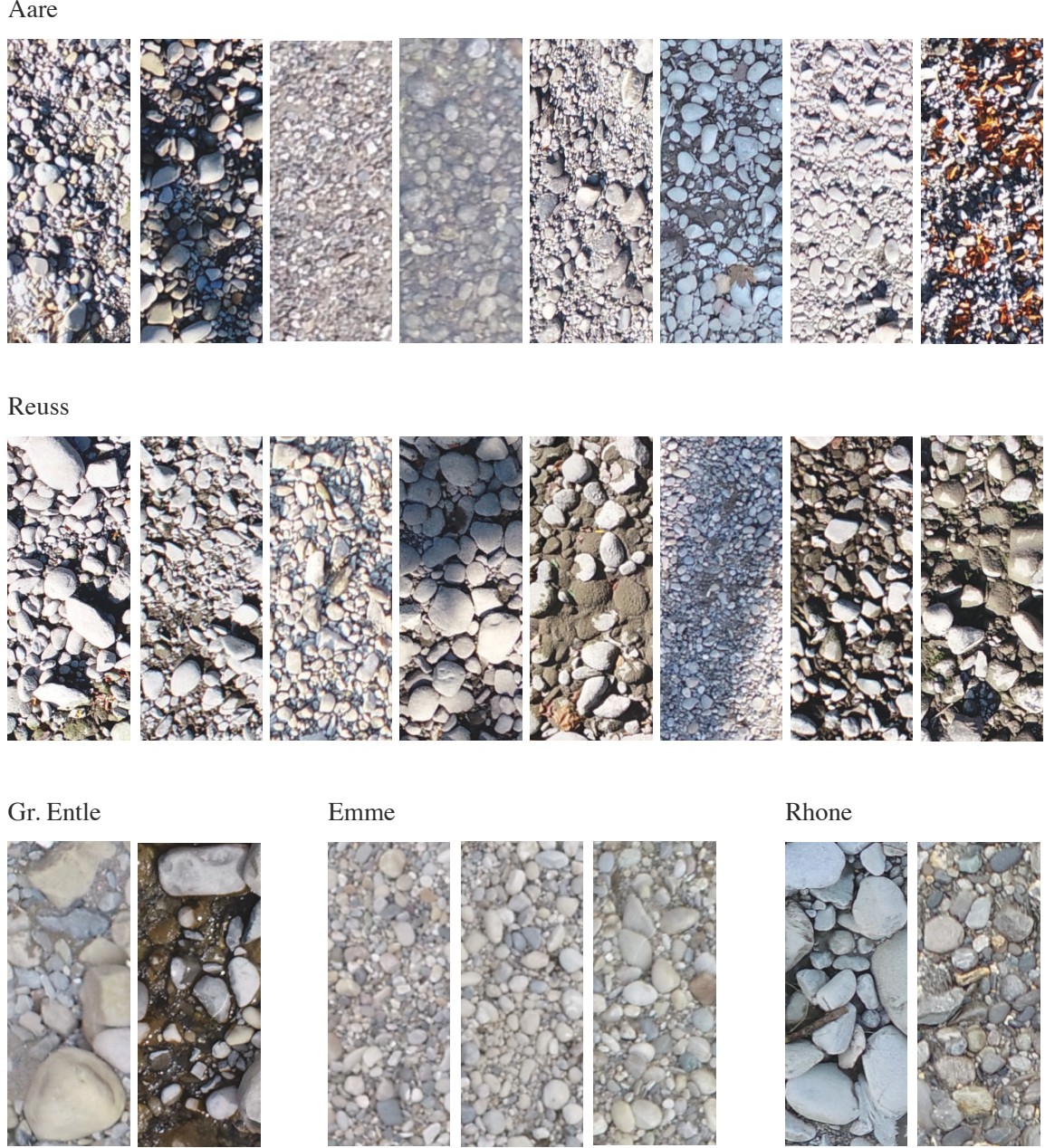

**Figure G1.** Tiles grouped by river name. The examples illustrate the variability between different bars on the along the same river.

**Appendix H:  Generalization performance per gravel bar**

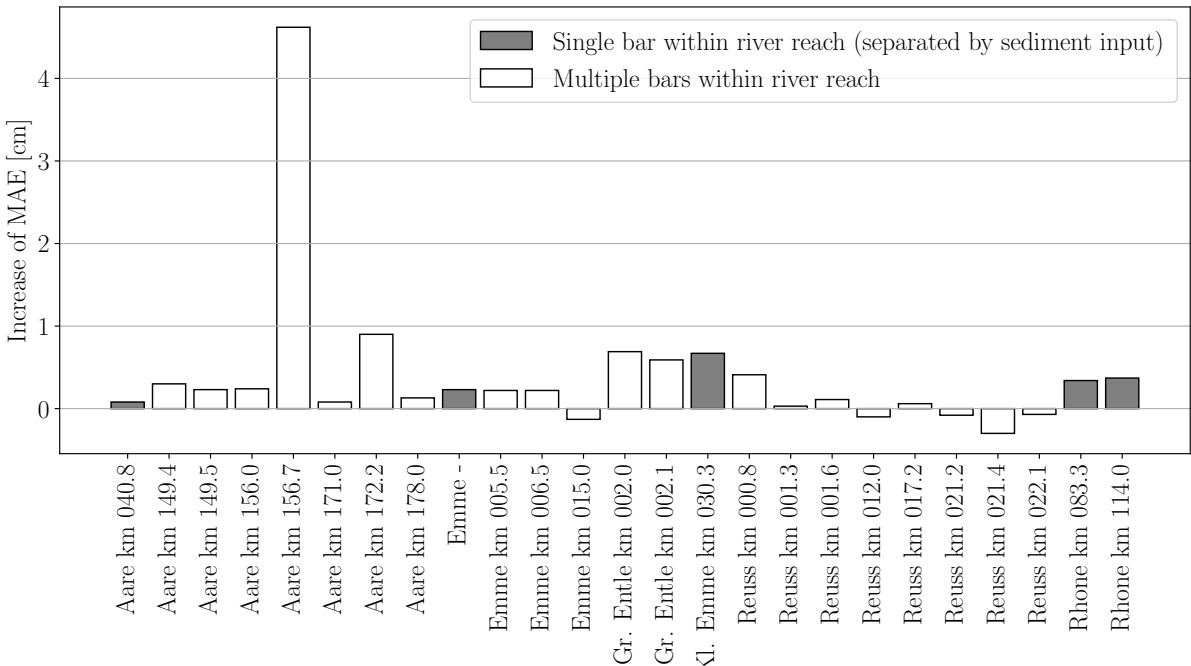

**Figure H1.** Generalization performance per gravel bar. Increase of mean absolute error (MAE) from the random cross-validation to the geographical generalization experiment for the *GRAINet* end-to-end $d_m$ regression. A positive value means that the generalization experiment has a higher error. Gravel bars on individual river reaches (i.e., separated by tributaries) are depicted in gray.