# Peer review of "GRAINet: Mapping grain size distributions in river beds from UAV images with convolutional neural networks"

_Hydrology and Earth System Sciences, 2020_

## Referee Comment (RC1) · Anonymous Referee #1 · 23 Jun 2020

General comments

The manuscript 'GRAINet: Mapping grain size distributions in river beds from UAV images with convolutional neural networks' meets the aspects, given on the HESS website at: https://www.hydrology-and-earth-system-sciences.net/peer_review/review_criteria.html

Specific comments

Section 3.1: what was the spatial resolution of produced orthophotos? One would assume that it was 0.25 m, but later in Section 4.1 you wrote that you resampled the image tiles (extracted from orthophotos) to uniform GSD of 0.25 m. Why this preprocessing step was necessary since PhotoScan enables user to set the specific spatial resolution when producing orthophotos?

Technical corrections

The numbering of Appendixes starts with 'B'. It would be more natural, to start the numbering by 'A'. Include the numbering of Appendixes when referring to them in the text (e.g. line 109: '... in the Appendix B...')

Consider unifying the expressions labelling and labeling, labelled and labelled (sometimes you use single 'L' sometimes double 'L'); 'bedload' and 'bed load'; 'analyse' and 'analyze' (and variations).

line 20: consider replacing 'As a conseguence' by 'Consequently' line 84: consider replacing 'photographs''' by 'images' line 171: insert ',' after For simplicity line 421: replace 'form' by 'from'

---

## Referee Comment (RC2) · Patrice Carbonneau (Referee) · 19 Jul 2020

Review of GRAINet

This paper develops a deep learning (DL) workflow to map grain size. The method is innovative because it fits the network directly to a distribution. The results seem extremely impressive and promising. But I have several major concerns that may need to be addressed before the work is ready for full publication.

Major issues

Drone Flights and Image resolution

[Figure]

The flight design for the method is rather inconvenient. Data is captured by imagery at an altitude of 10m with full 80% overlap for SfM. What the authors don't explicitly mention is that this leads to very long flight times to cover even small bars. This is not the norm. Many UAV users (and there are many of us) will read this paper wanting to apply the method to our fieldwork. But most drone acquisitions are done at 50+ meters. So in essence, the method as described here is a DL equivalent to Robotic Photosieving (cited in the work). The method would have much more impact if it could operate on the existing flight patterns (50+ meters flight altitudes) and show that you don't need the near-ground flights to get grain size distribution. The authors did try to look at the effect of resolution. But in my experience with image texture work, I have repeatedly found that downsampling an image digitally is not the same thing as acquiring the scene from a higher altitude. Such operations are approximations at best. As presented the method is an interesting alternative to Robotic Photosieving, but it doesn't have the transformative effect we have come to expect from Deep Learning methods. In section 5.45, the authors claim that it is novel to have dense grain size maps at scales of 1.25m by 0.5m. This is incorrect. Carbonneau et al (2004), which is actually cited here, produced grain size maps using image texture calculated on a 33x33 pixel window for 3cm imagery, the scale was therefore 1mx1m which is arguably very similar to the scale here. In 2005, the same group of Carbonneau et al showed that this could derive maps of D50 for en entire river of 80Km and presented grain size profiles based on several million image-based measurements of D50. The authors should then consult Woodget, Fyffe and Carbonneau (2018) in ESPL where we present a further image-bsaed method for particle sizing. For the drone th authors are using, operations at 50m AGL will deliver imagery at ∼2cm. This means that a texture based approach can deliver D50 estimates for sub-m2 patches. The authors need to choose their wording more carefully. The innovation here is that for each patch, GRAINet gives a distribution. But in terms of mean diameter, this has been possible for more than 15 years. Even in terms of the distribution, if a user applies the Robotic Photosieving method and spends as much time in near ground acquisition, then distributions can be

derived from a large number of images. We have found that the new PebbleCounts algorithm (Puriton and Bookhagen, this journal) is much faster and more accurate than Basegrain. This will get distributions for each image. This means that the progress of GRAINet is in processing time and perhaps better precision. But the issues with generalisation mean that users will need to train the network for new rivers. Ultimately, the real innovation potential here would be to show that this method works on a more 'standard' UAV orthomosaics. This would really position the paper as a major innovation on existing methods.

Network Architecture Description

The authors provide a qualitative description of their architecture, and a very detailed description of their loss functions, But the actual architectures they use remain very opaque. See below my point on code availability. At the very least the work needs a figure to detail the network architecture. This is very common in the DL literature. Why this network with residua blocks? Why not just a VGG16 or 19 model with a dense top terminating in a regression layer? Or any other model? Which libraries are used? Even this detail is lacking. It would also be helpful if the authors could show some of the activation maps in order to confirm that the network is 'seeing' reasonable elements that can conceivably lead to a regression to grain size distributions.

Training and Overfitting

In a wider perspective, this is a medium network with in excess of 1 million parameters. Since the classic Goodfellow textbook recommends 5000 samples per class in a classification problem, the sample size presented here does seems small even if I fully realise that much work went into it's production. But it then follows that the authors must establish that their network is well trained. As it stands, there is even a footnote saying it is overfit. Readers less familiar with DL might not realise it, but this is a very serious problem. The authors need to provide the reader with some crucial re-assurances and at least show a tuning plot as seen in figure 1 (from supplementary

information in Carbonneau et al 2020 in ESPL early view).

Validation

The authors maximise their small dataset and use k-fold cross-validation. This is fine to start. But what they call 'geographical cross-validation' is nothing else but bar-scale boot-strapping (sometimes called jack-knifing). It is not appropriate or reliable in this context. The main issue is that even of the authors hold out a whole bar, there remains samples from the same river. Given that river properties vary relatively slowly with geology and sometimes tributary inputs, the boot-strapped training samples will still have data that is very similar to the label data. So this is not a good test of generalisation. A much better approach to this would be to hold out an entire river. If this does not give good results, then the authors must sell this approach as river specific, and consider how much time a new user would need to spend in collecting and producing data to train a new GRAINet. As it stands, I think the results show that the network is not well trained, has too many parameters or too little training data.

Comparison to past work

I have no doubt that this deep learning approach will do better approaches than old texture-based approach, but the authors must be more direct in their discussion when the describe their outcomes. They must show that this is not just a new way of doing existing jobs, it represents real progress. And to do this they must cite errors from other work and clearly demonstrate that their results are better. This should include SedNet which is the rival CNN approach. And given that the title has the term UAV, they must be much more explicit in discussing their method in relation to other UAV (and airborne) particle sizing methods. Ideally, they should try to adapt the method so that it represents real progress with respect to other UAV methods in terms of time spent in acquisition, pre-processing and final data quality.

Code

Perhaps I am missing something but I cannot find the code associated to this work. If this is not a mistake on my part, I view this as a critical limitation of the work. As the authors surely know, the ethics of the deep learning community are strongly oriented towards the open source model. There are countless GitHub sites describing CNN architectures. If Google and Facebook can open-source Tensorflow and Torch, I cannot understand why the authors cannot make their code transparent. In the case of this work, the opaque architecture descriptions make this even more important. Scientists are increasingly criticised for a lack of transparency and something we can and should all do is release the code for such methodological contributions. I leave it to the Editor to decide how important these thoughts should be as this is ultimately a Copernicus policy decision. But I think the time where such methods could be unseen has passed.

Minor issues I have attached an annotated manuscript.

Overall recommendation This is an exciting method that potentially represents a major shift in our ability to measure grain sizes from UAVs. But the work needs major revision before full publication. Key issues:

1. Did the authors not conduct a more conventional UAV survey at higher altitudes when they acquired the data? It would really strengthen the paper if they could show that the method is applicable to existing UAV-derived orthomosaics of rivers sediments now routinely acquired by a large number of researchers. Do higher altitude images even have the required information content to infer distribution? 2. Provide a clearer description of their architecture along with a figure. Indicate libraries they are using. 3. Redo the 'geographic cross validation' to have a proper out-of-sample test set by setting aside entire rivers. 4. Improve and clarify the training procedure with an emphasis on showing that their re-trained models are not overfit. Unfortunately this may require the reprocessing of the data, it's not acceptable to have a footnote and say that your models are overfit, that means they are not ready for publication. 5. Provide a batter comparison with past work and be sure that literature cited in this work has been fully read. 6. Seriously consider a more transparent approach with open-source code.

[Figure]

Patrice Carbonneau July 2020

Please also note the supplement to this comment:
https://www.hydrol-earth-syst-sci-discuss.net/hess-2020-196/hess-2020-196-RC2-supplement.pdf

[Figure]

**Fig. 1.**

**Supplement:**

**GRAINet: Mapping grain size distributions in river beds from UAV images with convolutional neural networks**

Nico Lang[1], Andrea Irniger[2], Agnieszka Rozniak[1], Roni Hunziker[2], Jan Dirk Wegner[1], and Konrad Schindler[1]

[1]EcoVision Lab, Photogrammetry and Remote Sensing, ETH Zürich, Switzerland
[2]Hunziker, Zarn & Partner, Aarau, Switzerland

**Correspondence:** Nico Lang (nico.lang@geod.baug.ethz.ch)

**Abstract.** Grain size analysis is the key to understand the sediment dynamics of river systems. We propose *GRAINet*, a data-driven approach to analyze grain size distributions of entire gravel bars based on georeferenced UAV images. A convolutional neural network is trained to regress grain size distributions as well as the characteristic mean diameter from raw images. *GRAINet* allows the holistic analysis of entire gravel bars, resulting in *(i)* high-resolution maps of the spatial grain size distribution at large scale, and *(ii)* robust grading curves for entire gravel bars. To collect a large training dataset of 1,
 samples, we introduce *digital line sampling* as a new annotation strategy, following the widely applied line sampling analysis of Fehr (1987). Our evaluation on 25 gravel bars along 6 different rivers in Switzerland yields high accuracy: The resulting maps of mean diameters have a mean absolute error (M.?.) of 1.1 cm, with no bias. Robust grading curves for entire gravel bars can be extracted if representative training data is available. At the gravel bar level the MAE of the predicted mean diameter is even reduced to 0.3 cm. Extensive experiments were carried out to study the quality of the digital line samples, the generalization capability of *GRAINet* to new locations, the model performance w.r.t. human labeling noise, the limitations of the current model, and the potential of *GRAINet* to analyze images with low resolutions.

**1 Introduction**

Understanding the hydrological and geomorphological processes of rivers is crucial for their sustainable development, so as to mitigate the risk of extreme flood events and to preserve the biodiversity in aquatic habitats. However, the fluvial morphology of many streams is heavily affected by human activity and construction along the river. Gravel extractions, sediment retention basins in the upper catchments, hydro power plants, dams or channels reduce the bed load and lead to surface armouring, clogging of the bed, and latent erosion (Surian and Rinaldi, 2003; Simon and Rinaldi, 2006; Poeppl et al., 2017; Gregory, 2019). As a consequence, the natural alteration of the river bed is hindered, eventually deteriorating habitats and potential spawning grounds. Moreover, the process of bedload transport can cause bed or bank erosion, the destruction of engineering

**Summary of Comments on hess-2020-196.pdf**

**Page: 1**

Number: 1     Author: patca     Subject: Sticky Note     Date: 17/07/2020 14:11:55
remove the word 'large'. In the context of deep learning, 1400 samples is not large.

Number: 2     Author: patca     Subject: Sticky Note     Date: 17/07/2020 14:12:44
no size dependence on MAE?

structures (e.g., due to bridge scours) or increased flooding due to deposits in the channel that amplify the impact of severe floods (Badoux et al., 2014).

What makes modelling of fluvial morphology challenging are the mutual dependences between the flow field, grain size, its
25  movement and the geometry of the channel bed and banks. While channel shape and roughness define the flow field, the flow moves sediments — depending on their size — and the bed is altered by erosion and deposition. This mutually reinforcing system makes understanding channel form and process hard. Transport calculations in numerical models are thus still based on empirical formulas (Nelson et al., 2016).

One important key indicator for modelling sediment dynamics of a river system is the *grading curve* of the sediment.
30  Depending on the complexity of the model, the grain size distribution is either described by its characteristic diameters (e.g., the mean diameter $d_m$ defined by Meyer-Peter and Müller, 1948) or by the fractions of the grading curve (fractional transport, Habersack et al., 2011). The grain size of the river bed is crucial because it defines the roughness of the channel as well as the incipient motion of the sediment. Thus, knowledge of the grain size distribution is essential to specify flood protection measures, to asses bed stability, to classify aquatic habitats, and to evaluate geological deposits (Habersack et al., 2011).
35  Collecting the required calibration data to describe the composition of a river bed is time-consuming and costly, since it varies strongly along the downstream of a river (Surian, 2002) and even locally within individual gravel bars (Babej et al., 2016).

Traditional mechanical sieving to classify sediments (Krumbein et al., 1938) requires a substantial amount of skilled labour, and the whole process of digging, transport, and sieving is time-consuming, costly, and destructive. Consequently, it is rarely implemented in practice. A simplified, more efficient approach that collects sparse data samples in the field is the *line sampling*
40  analysis of Fehr (1987), the quasi-gold standard in practice today.[1] Yet, this approach is still very time-consuming and, worse, potentially inaccurate and subjective (Detert and Weitbrecht, 2012). Moreover, data collection requires physical access and cannot adequately sample inaccessible parts of the bed, such as gravel bar islands.

An obvious idea to accelerate data acquisition is to estimate grain size distribution from images. So-called *photo-sieving* methods that manually measure gravel sizes from ground level images (Adams, 1979; Ibbeken and Schleyer, 1986) were first
45  proposed in the late 1970s. Much research tried to automatically estimate grain size distributions from ground level images (Butler et al., 2001; Rubin, 2004; Graham et al., 2005; Verdú et al., 2005; Detert and Weitbrecht, 2012; Buscombe, 2013; Black et al., 2014; Spada et al., 2018; Buscombe, 2019). On the contrary, relatively little research has addressed mapping of grain sizes from images at larger scale (Carbonneau et al., 2004, 2005, 2018; Zettler-Mann and Fonstad, 2020), needed for practical impact.
50  Other researchers have proposed to analyze 3D-data acquired with terrestrial or airborne LiDAR, or through photogrammetric stereo matching (Brasington et al., 2012; Vázquez-Tarrío et al., 2017; Wu et al., 2018; Huang et al., 2018). However, working with 3D-data introduces much more overhead in data processing compared to 2D-imagery. Moreover, terrestrial data acquisition lacks flexibility and scalability, while airborne LiDAR remains costly (at least until it can be recorded with consumer-grade UAVs). Photogrammetric 3D-reconstruction may be limited by the ground sampling distance of the images,
55  leading to the absence of smaller grains.
* * *
[1]To the best of our knowledge, this includes at least the German-speaking countries: Switzerland, Germany, and Austria.

**Number: 1**     Author: patca     Subject: Sticky Note     Date: 17/07/2020 14:26:02

There is too much tendency for short paragraphs. Tighten this up.

**Number: 2**     Author: patca     Subject: Sticky Note     Date: 17/07/2020 14:26:16

Black et al was airborne work, should be in the next set

**Number: 3**     Author: patca     Subject: Sticky Note     Date: 17/07/2020 14:27:28

Mising paper by Woodget and Austrums, DeHaas et al (debris fans using UAV granulometry). Dugdale et al on Aerial Photosieving as well

**Number: 4**     Author: patca     Subject: Sticky Note     Date: 17/07/2020 14:29:30

I see your point but this does not seem accurate. The image ground sampling distance will also have an effect on any image-based DL method. But in SfM, the final resolution of the point cloud is never as high as the imagery. Classically, we'd assume the DEM needs at least 2x2 image pixels, but with SfM clouds, this relationship is not so universal.

[revised manuscript text omitted]

Number: 1     Author: patca     Subject: Sticky Note     Date: 17/07/2020 14:41:33
strictly speaking, the convolution operation is a dot product, not a scalar product.

Number: 2     Author: patca     Subject: Sticky Note     Date: 17/07/2020 14:43:29
In a paper on grain size, can you please avoid using d or D for parameters other than particle dimensions?

Number: 3     Author: patca     Subject: Sticky Note     Date: 17/07/2020 14:45:36
An you please give an example.  Readers amiliar with DL will associate the softmax activation with a classification output where you use the argmax function to determine the final class.  Why not just pass a linear activation function with the same number of nodes as samples in the distribution (eg 3 nodes for D16, D50, D84) ?

Number: 4     Author: patca     Subject: Sticky Note     Date: 17/07/2020 14:46:18
But still very large compared to your sample size. Given the footnote above about overfitting, I would be cautious.

Number: 5     Author: patca     Subject: Sticky Note     Date: 17/07/2020 14:47:10
Nee a figure for network architecture.  Pulbished code is mentionned, where is it published?

[revised manuscript text omitted]

Number: 1     Author: patca     Subject: Sticky Note     Date: 17/07/2020 14:57:18

I have no doubt that DL approaches will surpass texture-based approaches, but more discussion is needed here because the texture based approaches are quite suitable for the same data source. You need to repeat the error statistics here and cite some errors from texture based approaches to demonstrate that GRAINet is better.

Number: 2     Author: patca     Subject: Sticky Note     Date: 17/07/2020 14:57:47

And this is boot strapping. Any botstrapped saple will have training data from all rivers. It is not proper out-of-sample validation.

[revised manuscript text omitted]

---

## Referee Comment (RC3) · Anonymous Referee #3 · 1 Aug 2020

GENERAL COMMENTS

This work by Lang et al. shows an exciting perspective for collecting information about sediment size in streams with coarse bed material. A new approach that combines UAV images and convolutional neural network is proposed. I think that this work could represent a significant and novel contribution, although some key points should be carefully addressed in the revision of the manuscript.

My main concerns about the work are:

1. Introduction. This section is quite weak, it should be improved significantly. Some suggestions for improving this section are given below. L 15-28. This part is not very

useful. It would be more useful to focus on why grain size data are crucial (e.g. process understanding, modelling). L 38-42. Reference to traditional approaches is very poor. I would avoid the reference to Fehr (1987), maybe a good reference in the German-speaking countries but not worldwide (and in an international journal). I would suggest to look and refer to classical works by Church, Bunte, and many others. For instance, a look to Bunte and Abt (2001, USDA) would be very useful to put this work in the general context of sediment sampling in gravel-bed rivers. L 56. "….is more efficient than traditional field measurements…": I would say that automatic grain size is much less time consuming but it is also, commonly, less accurate. This should be pointed out since it is probably not obvious for readers who are not familiar with sediment sampling. Besides expanding the references considering previous works about sediment sampling, it could be useful to refer to works (e.g. Rice and Church, 2010, Sedimentology) that analyzed lateral and longitudinal variations of sediment size within a single bar. This would be useful to show the great potentials that GRAINet would offer for different purposes (e.g. sediment transport processes, morphodynamic and hydraulic modelling, ecological assessment).

2. Ground truth (see section 3.3). Is this really a ground truth? These measurements of grains are obtained from images not from direct measurements. I understand that this can be the way for training the model, but I would not say that these are ground truth…two different things! This is a key point that should be carefully addressed: the term ground truth is used widely throughout the manuscript.

3. Comparison with field measurements. This is a weak part of the work (see also my previous comment). (i) How field measurements were carried out should be explained in detail (in the Method section). (ii) A better comparison with digital line samples should be carried out: I do not agree that "…overall, no bias exists between the field measurements and the digital line samples" (L 343-344; figure 6). At least for those field measurements of known location, it would be crucial to show the real difference with digital line samples (e.g. if dm is 3-4 cm, difference of 1 cm or more is quite

significant). I think that this is a crucial part of the work that needs to be improved. It is crucial to show how close, or not, are data obtained by GRAINet to those obtained by field measurements.

4. "...Our CNN-based approach makes it possible to robustly estimate grain size distributions and characteristic mean diameters from raw images..." (L 507-508). This conclusion is not sufficiently supported by data (see my previous comments, 2 and 3).

5. Manual component of GRAINet. I have two comments about this aspect. (i) "...Lastly, one could potentially support the still necessary manual annotation process using training-free image processing tools, such as the open source software BASEGRAIN..." (L 553-554). This is a good point that would require further discussion. It is mentioned only in the last section ("Conclusions"). (ii) Comparison with human performance (section 5.4.4). Errors are not so small, see figure 15: further discussion would useful here. It would be useful to clarify better how much the manual component affects the overall performance of GRAINet.

6. Discussion ("Advantages and limitations of the approach"). What about the presence of fine material? How the presence of fine material would affect the results obtained with GRAINet? This is something that should be discussed.

SPECIFIC COMMENTS

Section 5.1. It could be moved in the Method section.

L 355. Quantitative analyses were carried out in the field, see for instance Wohl et al. (1996, WRR).

L 502. "...We believe that, in principle, GRAINet could even be used to process airborne imagery from country-wide flight campaigns...". I am quite skeptical about this statement. Commonly country-wide flights have spatial resolution of 20-30 cm (or lower resolution): such resolution seems to be too low to obtain reliable results (see figure 19 and 20).

L 540-541. "...to successfully replace the gold standard line sampling in the field (Fehr, 1987)...": as I pointed out in a previous comment, this is not a good reference for an international audience and journal.

---

## Referee Comment (RC4) · Patrice Carbonneau (Referee) · 3 Aug 2020

Since this is meant as a discussion, I'd like to point out that the issue of ground-truthing in photosieving has been popping up in the reviews of such work for decades. The core question is what are you measuring? Anybody that uses photosieving must be made aware that they should not expect the same grain size that they would obtain from a bulk sample. It is not even as simple as equating it to a Wolman count because armoring and embedding confuse what you can see from a nadir photo.

But there is work reporting such errors. In past papers, the solution has usually been to mention this body of work and recognise the caveats above. I think this could go in

the same direction as my own comments along the lines that this paper must do more to discuss errors of past photosieving work in relation to the results presented here.

Patrice Carbonneau

---

## Author Comment (AC1) · 26 Aug 2020

We are glad that the reviewer finds our work relevant for the HESS journal in terms of scientific significance and quality. Thank you for reviewing our paper and pointing out the technical corrections. We will consider these in our revised version.

Regarding your open question:

*"Why this preprocessing step was necessary since PhotoScan enables user to set the specific spatial resolution when producing orthophotos?"*

You are right, we could have fixed the resolution when generating the orthophotos. In

fact the ground sampling distance (GSD) of all orthophotos is not far from 0.25 cm. We preferred to keep the best possible GSD to support the manual data annotation, then resample to exactly 0.25 cm for the CNN. We see this as a minor technical detail.

Either way, a uniform resampling for the automatic processing is beneficial, as it allows the CNN to implicitly learn the absolute scale, and consequently to predict the frequency of absolute, metric object sizes without any additional information about scale or resolution. Another advantage of (any) constant resolution is that it simplifies the learning task, since the visual features need not be scale-invariant (which in our application would be unnecessary and could even be a disadvantage, introducing scale ambiguities).

––––––––––––––––––––––––––––––––––

---

## Author Comment (AC2) · 26 Aug 2020

Dear Patrice Carbonneau,

Thank you for your thoughtful feedback. It is motivating that you find our work innovative and the results extremely impressive and promising. Thank you for pointing out technical details, which we will incorporate in the revised version. We will refer to the additional related work, where appropriate, in the revised paper. We address your discussion points in the sections below.

**Drone Flights and Image resolution**

[Figure]

*"The method would have much more impact if it could operate on the existing flight patterns (50+ meters flight altitudes) and show that you don't need the near-ground flights to get grain size distribution."*

The choice of image resolution was driven by the requirement that a human annotator should be able to reliably label individual grains down to 1 cm in size. We found that a ground sampling distance of 0.25 cm is the limit for this task. To achieve that resolution, the specific UAV used in this study flew 10 m above the ground. However, the flying height depends only on the image magnification, flying at higher altitude is perfectly possible with a suitably chosen camera payload (sensor and lens). Hence, we do not see the input resolution of 0.25 cm as a serious limitation of our proposed approach, especially given the fast progress of continuously improving and ever cheaper hardware.

*"The authors did try to look at the effect of resolution. But in my experience with image texture work, I have repeatedly found that downsampling an image digitally is not the same thing as acquiring the scene from a higher altitude."*

We study the effect of lower input resolutions on the performance of the grain size estimation by downsampling the high-res images (with appropriate smoothing to avoid aliasing effects). To our knowledge this is the best practice and widely used e.g. in the image analysis and computer vision community to study super-resolution tasks. Our study shows that the performance of the CNN is not affected up to 2 cm ground sampling distance (downsampling factor 8), which would be the comparable resolution achieved at 80 m flying altitude with the particular low-cost UAV used in this work. We are confident that, although our synthetic downsampling may indeed miss subtle optical effects, actual images taken with 1 cm GSD would perform at least as well as simulated ones with 2 cm. Since we do not have high-altitude orthophotos for the investigated gravel bars, this is the best study we can provide.

*"The authors need to choose their wording more carefully. The innovation here is that*

*for each patch, GRAINet gives a distribution."*

We agree and will be more precise in describing the novelty with respect to your work. To summarize, our presented approach has the following contributions:

- End-to-end estimation of the grain size distribution at particular locations over an area of 1.25 m x 0.5 m

- Robust grain size distribution for entire gravel bars.

- Overall performance is invariant to the image resolution up to 2 cm ground sampling distance.

- Generic approach that also allows to map particular grain size metrics like the mean diameter with the same model architecture.

- Mapping of mean diameters <1.5 cm

**Training and Overfitting**

*"As it stands, there is even a footnote saying it is overfit."*

This seems to be a misunderstanding. To clarify, the footnote number 2 on page 4, referring to line 94, is talking about the related work by Buscombe (2019), where it appears that model selection and performance evaluation were done with the same data, which typically leads to an overfit with overly optimistic results. "Model selection" in this context means choosing at which iteration of the learning sequence the network parameters are read out. That hyper-parameter must be tuned on a validation set, without looking at the performance on the actual test data (see also Goodfellow et al. (2016), Chapter 5, section 5.3). Besides sanity-checking the training, a main purpose of monitoring the validation loss is to detect when the relatively best parameters have been found, as overfitting to the training data begins.
Contrary to our understanding of Buscombe's work, we do use the correct procedure (line 323 in the paper), i.e., our model is not overfitted and the reported performance is an unbiased estimator of the expected performance on unseen images.

*"...the authors must establish that their network is well trained."*

We carried out a range of careful experiments to evaluate the performance of the network, and present its performance change w.r.t. random training-test splits and w.r.t. particular training-test gravel bars. As mentioned in line 471, the latter is coupled with particular training-test imaging conditions, since each gravel bar was recorded individually, with specific imaging conditions. In our opinion this is a rather comprehensive and transparent form of evaluation. We also tried to explicitly discuss both advantages and weaknesses. If you believe there is a better way to assess model training (with the available data), perhaps you could suggest a concrete experiment?

**Validation**

*"But what they call 'geographical cross-validation' is nothing else but bar-scale boot-strapping (sometimes called jack-knifing). It is not appropriate or reliable in this context. The main issue is that even if the authors hold out a whole bar, there remain samples from the same river. Given that river properties vary relatively slowly with geology and sometimes tributary inputs, the boot-strapped training samples will still have data that is very similar to the label data. So this is not a good test of generalisation. A much better approach to this would be to hold out an entire river."*

We respectfully disagree, but we agree that this is a good discussion point and needs to be clarified in the paper.
In some geographical conditions the river properties may vary slowly, but the characteristics of the investigated bars can hardly be grouped by the river name. The relevant statistics of the investigated gravel bars are presented in Table B1 in the paper. The characteristics of the investigated gravel bars from the same river are diverse, both

quantitatively (mean Dm, Dm range in Table B1) and qualitatively (see the attached Figure 1, where tiles are grouped by river name). Not only is this due to the distance between the bars, but also due to the changing slope and the varying river bed widths in mountain environments (Reuss, Aare, Emme). Furthermore, the bars are geographically separated through tributaries (Aare, Rhone), leading to a drastic increase in the catchment areas between the bars. E.g., at the river Rhone the catchment area is more than doubled from 982 km$^2$ (at km 083.3) to 2485 km$^2$ (at km 114.0). On the other hand the sediment transport is affected by dams at the rivers Aare, Rhone, and Reuss. Finally, the characteristics may also be artificially altered, as it is nowadays common in Central Europe to replenish gravels of 2-3 cm to create spawning grounds for fish. For instance, the grain size distribution at the bar Reuss km 022.1 (and probably km 012.0) is very likely affected by such a targeted replenishment of sediment.

Hence, we are convinced that the presented hold-one-bar-out experiment is a valid setup to evaluate the generalization of the approach to new locations. In this setup, the data is exploited best, allowing the CNN to learn features invariant to the imaging conditions by providing 24 different orthophotos in each experiment. If we would hold out, e.g., the whole river Aare, not only the number of training samples would be substantially reduced, but also the diversity of imaging conditions.

In fact, within our experimental setup we already present one hold-one-river-out experiment for the river Kl. Emme, from which only one bar is included in our dataset. Even though this bar contains the largest number of digital line samples, its estimated grain size distribution fits rather well in the geographical cross-validation experiment (see Figure 17). The observed performance drop between the random and geographical cross-validation experiment for individual bars in Figure 18 is rather explained by coarse gravel bars with a large mean Dm and a wide Dm range. Seeing bars from the same river during training and testing does not seem to have an effect (for instance, Aare).

It is also important to keep in mind that data-driven approaches, like the one proposed, will only give reasonable estimates if the test data approximately matches the training

data distribution. It will not perform well for out-of-distribution (OOD) samples. Detecting such OOD samples is an unsolved problem and an active research direction.

*"As it stands, I think the results show that the network is not well trained, has too many parameters or too little training data."*

We do not understand this statement. Since no further explanation is given, we cannot properly address it. We can only hypothesize that this claim is based on the misunderstanding of footnote 2, which we have discussed above.
In fact, our experimental setup quite clearly reveals in which specific cases the limited and unbalanced training data causes performance issues. Namely, our results degrade for coarse gravel bars, which is linked to the fact that only 14% of the training data have Dm > 10 cm (see lines 515-522 in the paper).

**Network Architecture Description**

*"Why this network with residual blocks?"*

After their introduction by He et al. (2016), CNNs with residual blocks quickly became the preferred design for deep image interpretation models (including many successors that no longer have the "residual" in their names). Residual connections improve gradient flow during training, because they provide shortcuts that mitigate the vanishing gradient problem. We refer to He et al. (2016), Veit et al. (2016) for more details. The model depth and capacity is empirically calibrated on the validation data for best performance on the grain size estimation task. In line with recent literature, our view is that a well-chosen loss function that matches the application objective is key, and more important than the specific choice of CNN architecture. Well-proven, contemporary architectures are rather interchangeable, and the right depth (capacity) depends on the specific task and dataset size. If a larger training set were available, an even deeper network might well perform better.

*"At the very least the work needs a figure to detail the network architecture."*

We will add the illustration of the network in the revised paper (see attached Figure 2).

*"It would also be helpful if the authors could show some of the activation maps in order to confirm that the network is 'seeing' reasonable elements that can conceivably lead to a regression to grain size distributions."*

Thank you for the suggestion, we will add examples of activation maps to the revised paper (see attached Figure 3). We show the activation maps after the last convolutional layer, before global average pooling. Hence, each of the 21 maps corresponds to a specific bin of grain sizes, with bin 0 for the smallest grains and bin 20 for the largest ones. Light colours are low activations, darker red denotes higher activations. To harmonize the activations to a common scale [0, 1] for visualisation, we pass the maps through a softmax over bins. The activation maps intuitively make sense, with smaller grains activating the corresponding, lower bin numbers.

**Comparison to past work**

*"They must show that this is not just a new way of doing existing jobs, it represents real progress. And to do this they must cite errors from other work and clearly demonstrate that their results are better. This should include SedNet which is the rival CNN approach. And given that the title has the term UAV, they must be much more explicit in discussing their method in relation to other UAV (and airborne) particle sizing methods. Ideally, they should try to adapt the method so that it represents real progress with respect to other UAV methods in terms of time spent in acquisition, pre-processing and final data quality."*

We discuss drawbacks of existing methods in the paper (line 56-61, line 85, line 94, line 100). The SediNet approach by Buscombe (2019) focuses on a different application (clean sediment and sand samples) and the experimental flaw discussed above (c.f. footnote 2 in the paper) does not allow for a meaningful comparison. To our knowledge ours is the first work that evaluates grain size estimation over entire gravel

bars in river beds, and we are not aware of a comparable study regarding geographical generalization.

Due to the end-to-end learning, the proposed CNN is able to extract global features that are informative about grain size beyond the sensitivity of human photo-interpretation as well as traditional photosieving that relies on local image gradients to delineate individual grains. Even the latest work of Purinton and Bookhagen (2019) can only detect individual grains that have a b-axis 20x the ground sampling distance. Also previous statistical approaches (Carbonneau 2004) are limited by the input resolution and can only predict D50 down to 3 cm, so we do believe that our approach overcomes some of the limitations of prior art.

Moreover, the CNN is a generic learning machine and can be seen as a "Swiss army knife" for grain size characterisation from images. It can be used to directly predict the full grain size distribution at a specific location, but the same architecture can also be trained to directly predict other desired grain size metrics derived from the distribution. A further advantage over most competitors, with large practical potential, is that our method scales to entire gravel bars, resulting in detailed, high-resolution maps and more robust spatial aggregations.

Obviously, creating a large, manually labelled training dataset is time-consuming, a property our CNN shares with other supervised machine learning methods. However, at test time the proposed approach requires no parameter tuning by the user, a considerable advantage for large-scale applications, where traditional image processing pipelines struggle, since they are fairly sensitive to varying imaging conditions.

**Code**

The project is implemented using Keras, with Tensorflow as a backend. The code and a demo will be published together with the paper. Unfortunately, we are unable to distribute the full dataset for commercial reasons, as it is owned by a private company (who also created it at their own cost).

**References**

Buscombe, D. (2020). SediNet: A configurable deep learning model for mixed qualitative and quantitative optical granulometry. Earth Surface Processes and Landforms, 45(3), 638-651.

Goodfellow, I., Bengio, Y., and Courville, A. (2016). Deep Learning, MIT Press, http://www.deeplearningbook.org. He, K., Zhang, X., Ren, S., Sun, J. (2016). Deep residual learning for image recognition. In Proceedings of the IEEE Conference on Computer Vision and Pattern Recognition (pp. 770-778).

Veit, A., Wilber, M. J., and Belongie, S. (2016). Residual networks behave like ensembles of relatively shallow networks. In Advances in Neural Information Processing Systems (pp. 550-558).

Purinton, B. and Bookhagen, B. (2019). Introducing PebbleCounts: a grain-sizing tool for photo surveys of dynamic gravel-bed rivers. Earth Surface Dynamics, 7(3).

[Figure]

[Figure]

**Fig. 1.** Tiles grouped by river name

[Figure]

**Fig. 2.** Illustration of the CNN architecture

**Fig. 3.** Activation maps after the last convolutional layer for two examples. Each of 21 maps corresponds to a specific bin of grain sizes, where bin 0 belongs to the smallest, bin 20 to the largest grains.

---

## Author Comment (AC3) · 26 Aug 2020

We thank the reviewer for his insightful and positive feedback. We are happy that our work is seen as showing a new exciting perspective, and as a significant and novel contribution. Thank you for the thoughtful inputs and the hints to the useful related work that will strengthen the motivation for our work. We address the major points below and will consider them for the revised version of the paper.

*"I would avoid the reference to Fehr (1987)"*

The main purpose of the paper is to propose and evaluate a new method to estimate

grain size distribution from raw UAV images, using convolutional neural networks. We see that the standard methodology to measure grain sizes in the field may vary between countries. Fortunately, there is no tight coupling, the CNN is agnostic and will learn to replicate the outcome of any consistent procedure for creating ground truth grain size distribution samples. The annotation strategy can easily be exchanged and the network retrained to estimate grain size distribution according to a different National or regional standard (e.g., grid sampling). However, since the line sampling by Fehr continues to be the standard field method in the German speaking world, we found our digital line sampling to be most efficient, while providing representative reference data with respect to the "gold standard" of line sampling in the field.

**Introduction**

Thank you for the inputs. We will incorporate the related work and will revise the introduction to clarify "why grain size data are crucial". We already describe the importance of grain size in line 29 and following.

*"L 15-28. This part is not very useful."*

In our opinion it is important to give a broad motivation for our research and explain its relevance for society. If the reviewer (respectively, editor) insists we will of course remove the paragraph, but we would much prefer to keep it.

*"I would say that automatic grain size is much less time consuming but it is also, commonly, less accurate."*

We agree and mention that the automatic grain size estimation from images is still limited in terms of accuracy when it comes to large scale applications (line 56 in the paper).

**Ground truth**

*"Is this really a ground truth? These measurements of grains are obtained from images not from direct measurements. I understand that this can be the way for training the model, but I would not say that these are ground truth."*

In machine learning terminology, the term "ground truth" refers to the data that is used to train and evaluate the model (being the upper bound the model can ever reach if it manages to perfectly replicate the annotations). To clarify the terminology, we explicitly declare in line 343, that the term "ground truth" refers to the digital line samples. In the revision we will clarify this earlier in the paper in section 3.3, to avoid any possible confusion.

**Comparison with field measurements**

*"How field measurements were carried out should be explained in detail (in the Method section)."*

We will explain that field samples were measured according to line sampling proposed by Fehr (1987). But we feel it would be too much to add another section in the methodology, given that this is not the focus of the paper, and a widely used standard practice.

*"At least for those field measurements of known location, it would be crucial to show the real difference with digital line samples."*

As mentioned in line 336, the location of the field measurements is not known exactly, because the field measurements were originally recorded for other purposes, within other projects. The best we can do with the available data is to evaluate the bar-level agreement between (independent) field work and our digital line sampling approach. For the training and evaluation of the CNN we consider the digital line samples to be representative (line 344). We do agree that a comparison between digital line samples and geolocalized field samples would be interesting, but since this data is not available, we cannot provide it in this paper.

*"A better comparison with digital line samples should be carried out: I do not agree that ". . .overall, no bias exists between the field measurements and the digital line samples" (L 343-344; figure 6)."*

To better assess the agreement between the Dm derived from field and digital line samples at the bar level, we compute the overall bias across the 22 bars with available field data (see Figure 6). The mean error (bias) amounts to -0.3 cm, which means that the digital Dm is on average slightly lower than the Dm derived from field samples. The mean absolute error is 0.9 cm. The reviewer will no doubt agree that field samples are unavoidably affected by the selected location (line 341) and also by operator bias (Wohl 1996).

Hence, we still conclude that within reasonable expectations the digital line samples are in good agreement with field samples and constitute representative training data.

**Robust estimation**

Figure 11 shows the estimated grain size distribution for entire gravel bars in comparison to the ground truth data. Given appropriate training data, the model estimates the grain size distributions very accurately. Due to the inherent regularisation, deep learning models tend to be fairly robust.

**Manual component of the presented approach**

Semi-automatic image labelling might be an alternative way to speed up the annotation process, but one would have to carefully avoid systematic algorithmic biases in the semi-automatic procedure, otherwise the CNN will almost certainly learn to faithfully reproduce those biases. Similarly, systematic behaviours of specific annotators may also be learned by the model. Ideally, training data should thus be generated by different annotators with comparable (preferably high) skill. Independent of possible biases, the automatic approach handles all samples consistently and allows for unbiased monitoring over long times, as there is no variation due to changed operators (Wohl et al.,

1996).

*"Comparison with human performance (section 5.4.4). Errors are not so small, see figure 15"*

The vertical axis label in Figure 15 seems to have been mangled during pdf generation. To clarify, the Y-axis in Figure 15 corresponds to the Dm, not the error. We compare the mean diameter (Dm) estimated by the CNN to the Dm from multiple annotations by different operators. This comparison requires a lot of manual labour (repeated, independent annotation by different people), thus only a small number of samples could be processed, which is not ideal for statistical analysis. However, it still gives a feeling for the human variation in the annotation process, with an average standard deviation across different operators of 0.5 cm for Dm. The max standard deviation from repeated annotation is 2.0 cm.
We correct a small mistake in line 454: The standard deviation of the labels (0.5 cm) should be compared with the root mean squared error (RMSE) of the model predictions, not with the mean absolute error MAE. Hence, regressing Dm with GRAINet yields RMSE=1.7 cm, from which 29% can be explained by the label noise in the test data."

**Presence of fine material**

We have already included two examples in Figure 8 b) and c) that show that the network can make robust predictions even in the case of slight disturbances caused by colmation through fine material, given such samples are provided during training.

**References**

Wohl, E. E., Anthony, D. J., Madsen, S. W., and Thompson, D. M. (1996). A comparison of surface sampling methods for coarse fluvial sediments. Water Resources Research, 32(10), 3219-3226.

---

## Author Response (AR1)

**GRAINet review answers**

**Major changes in the revised version**

1. Revised introduction, included additional related work (RC3, RC2)
2. Added CNN architecture illustration (RC2)
3. Added results and visualizations of the learned attention maps (RC2)
4. Extend comparison between digital and field line samples (RC3)
5. Discussion on *Manual component of the presented approach (RC3)*
6. Discussion on *Geographical cross-validation (RC2)*
7. Discussion on *Comparison to past work* (RC2)

All changes in the revised paper are highlighted in blue.
Below is a systematic list of our answers.

**RC1**

*"Why this preprocessing step was necessary since PhotoScan enables user to set the specific spatial resolution when producing orthophotos?"*

This is correct, we could have fixed the resolution when generating the orthophotos. In fact the ground sampling distance (GSD) of all orthophotos is not far from 0.25 cm. We preferred to keep the best possible GSD to support the manual data annotation, then resample to exactly 0.25 cm for the CNN. We see this as a minor technical detail.

**Minor comments are addressed in the paper.**

**RC2**

**Drone Flights and Image resolution**

*"The method would have much more impact if it could operate on the existing flight patterns (50+ meters flight altitudes) and show that you don't need the near-ground flights to get grain size distribution."*

The choice of image resolution was driven by the requirement that a human annotator should be able to reliably label individual grains down to 1 cm in size. We found that a ground sampling distance of 0.25 cm is the limit for this task. To achieve that resolution, the specific UAV used in this study flew 10 m above the ground. However, the flying height depends only on the image magnification, flying at higher altitude is perfectly possible with a suitably chosen camera payload (sensor and lens). Hence, we do not see the input

resolution of 0.25 cm as a serious limitation of our proposed approach, especially given the fast progress of continuously improving and ever cheaper hardware.

*"The authors did try to look at the effect of resolution. But in my experience with image texture work, I have repeatedly found that downsampling an image digitally is not the same thing as acquiring the scene from a higher altitude."*

We study the effect of lower input resolutions on the performance of the grain size estimation by downsampling the high-res images (with appropriate smoothing to avoid aliasing effects) in section 5.5. To our knowledge this is the best practice and widely used e.g. in the image analysis and computer vision community to study super-resolution tasks. Our study shows that the performance of the CNN is not affected up to 2 cm ground sampling distance (downsampling factor 8), which would be the comparable resolution achieved at 80 m flying altitude with the particular low-cost UAV used in this work. We are confident that, although our synthetic downsampling may indeed miss subtle optical effects, actual images taken with 1 cm GSD would perform at least as well as simulated ones with 2 cm. Since we do not have high-altitude orthophotos for the investigated gravel bars, this is the best study we can provide.

*"The authors need to choose their wording more carefully. The innovation here is that for each patch, GRAINet gives a distribution."*

**Addressed in the paper: lines 82-87, line 492 (blue text)**

**Network Architecture Description**

*"Why this network with residual blocks?"*

After their introduction by He et al. (2016), CNNs with residual blocks quickly became the preferred design for deep image interpretation models (including many successors that no longer have the "residual" in their names). Residual connections improve gradient flow during training, because they provide shortcuts that mitigate the vanishing gradient problem. We refer to He et al. (2016), Veit et al. (2016) for more details. The model depth and capacity is empirically calibrated on the validation data for best performance on the grain size estimation task. In line with recent literature, our view is that a well-chosen loss function that matches the application objective is key, and more important than the specific choice of CNN architecture. Well-proven, contemporary architectures are rather interchangeable, and the right depth (capacity) depends on the specific task and dataset size. If a larger training set were available, an even deeper network might well perform better.

*"At the very least the work needs a figure to detail the network architecture."*

**Addressed in the paper: line 206, page 35 Figure B1**

*"It would also be helpful if the authors could show some of the activation maps in order to confirm that the network is 'seeing' reasonable elements that can conceivably lead to a regression to grain size distributions."*

**Addressed in the paper: page 19 lines 425-433, Figure 10**

**Training and Overfitting**

*"As it stands, there is even a footnote saying it is overfit."*

This seems to be a misunderstanding. To clarify, the footnote number 2 on page 5, referring to line 108, is talking about the related work by Buscombe (2019), where it appears that model selection and performance evaluation were done with the same data, which typically leads to an overfit with overly optimistic results. "Model selection" in this context means choosing at which iteration of the learning sequence the network parameters are read out. That hyper-parameter must be tuned on a validation set, without looking at the performance on the actual test data (see also Goodfellow et al. (2016), Chapter 5, section 5.3). Besides sanity-checking the training, a main purpose of monitoring the validation loss is to detect when the relatively best parameters have been found, as overfitting to the training data begins.

Contrary to our understanding of Buscombe's work, we do use the correct procedure (line 326 in the paper), i.e., our model is *not* overfitted and the reported performance is an unbiased estimator of the expected performance on unseen images.

**To avoid any confusions, we cite Buscombe (2019) explicitly in this footnote 2 on page 5.**

*"…the authors must establish that their network is well trained."*

We carried out a range of careful experiments to evaluate the performance of the network, and present its performance change w.r.t. random training-test splits and w.r.t. particular training-test gravel bars. In our opinion this is a rather comprehensive and transparent form of evaluation. We also tried to explicitly discuss both advantages and weaknesses.

**Validation**

*"But what they call 'geographical cross-validation' is nothing else but bar-scale boot-strapping (sometimes called jack-knifing). It is not appropriate or reliable in this context. The main issue is that even if the authors hold out a whole bar, there remain samples from the same river. Given that river properties vary relatively slowly with geology and sometimes tributary inputs, the boot-strapped training samples will still have data that is very similar to the label data. So this is not a good test of generalisation. A much better approach to this would be to hold out an entire river."*

*"As it stands, I think the results show that the network is not well trained, has too many parameters or too little training data."*

We do not understand this statement. Since no further explanation is given, we cannot properly address it. We can only hypothesize that this claim is based on the misunderstanding of footnote 2, which we have discussed above.

In fact, our experimental setup quite clearly reveals in which specific cases the limited and unbalanced training data causes performance issues. Namely, our results degrade for coarse gravel bars, which is linked to the fact that only 14% of the training data have Dm > 10 cm (see page 33 lines 616-621 in the paper).

**Comparison to past work**

*"They must show that this is not just a new way of doing existing jobs, it represents real progress. And to do this they must cite errors from other work and clearly demonstrate that their results are better. This should include SedNet which is the rival CNN approach. And given that the title has the term UAV, they must be much more explicit in discussing their method in relation to other UAV (and airborne) particle sizing methods. Ideally, they should try to adapt the method so that it represents real progress with respect to other UAV methods in terms of time spent in acquisition, pre-processing and final data quality."*

**Code**

The project is implemented using Keras, with Tensorflow as a backend. The code and a demo on a subset of the data is published here: https://github.com/langnico/GRAINet
Unfortunately, we are unable to distribute the full dataset for commercial reasons, as it is owned by a private company (who also created it at their own cost).
Nevertheless, the dataset may be requested for research purposes by directly contacting andrea.irniger@hzp.ch. (see revised paper "Code and data availability")

**References**

Buscombe, D. (2020). SediNet: A configurable deep learning model for mixed qualitative and quantitative optical granulometry. Earth Surface Processes and Landforms, 45(3), 638-651.

Goodfellow, I., Bengio, Y., and Courville, A. (2016). Deep Learning, MIT Press, http://www.deeplearningbook.org.

He, K., Zhang, X., Ren, S., & Sun, J. (2016). Deep residual learning for image recognition. In Proceedings of the IEEE Conference on Computer Vision and Pattern Recognition (pp. 770-778).

Veit, A., Wilber, M. J., & Belongie, S. (2016). Residual networks behave like ensembles of relatively shallow networks. In *Advances in Neural Information Processing Systems* (pp. 550-558).

Purinton, B., & Bookhagen, B. (2019). Introducing PebbleCounts: a grain-sizing tool for photo surveys of dynamic gravel-bed rivers. *Earth Surface Dynamics*, 7(3).

**Minor comments in the annotated pdf (supplement) are addressed in the paper.**

**RC3**

*"I would avoid the reference to Fehr (1987)"*

**Addressed in the paper: line 40**

The main purpose of the paper is to propose and evaluate a new method to estimate grain size distribution from raw UAV images, using convolutional neural networks. We see that the standard methodology to measure grain sizes in the field may vary between countries. Fortunately, there is no tight coupling, the CNN is agnostic and will learn to replicate the outcome of any consistent procedure for creating ground truth grain size distribution samples. The annotation strategy can easily be exchanged and the network retrained to estimate grain size distribution according to a different National or regional standard (e.g., grid sampling). However, since the line sampling by Fehr continues to be the standard field method in the German speaking world, we found our digital line sampling to be most efficient, while providing representative reference data with respect to the "gold standard" of line sampling in the field.

**Introduction**

*"Reference to traditional approaches is very poor"*

**Addressed in the paper: lines, 36, 40, 52**

*"L 15-28. This part is not very useful."*

In our opinion it is important to give a broad motivation for our research and explain its relevance for society. If the reviewer (respectively, editor) insists we will of course remove the paragraph, but we would much prefer to keep it.

*"I would say that automatic grain size is much less time consuming but it is also, commonly, less accurate."*

We agree and mention that the automatic grain size estimation from images is still limited in terms of accuracy when it comes to large scale applications (line 61 in the paper).

**Ground truth**

*"Is this really a ground truth? These measurements of grains are obtained from images not from direct measurements. I understand that this can be the way for training the model, but I would not say that these are ground truth."*

**Addressed in the paper: line 166**

**Comparison with field measurements**

*"How field measurements were carried out should be explained in detail (in the Method section)."*

We explain that field samples were measured according to the line sampling proposed by Fehr (1987) (page 15 line 354). But we feel it would be too much to add another section in the methodology, given that this is not the focus of the paper, and a widely used standard practice.

*"At least for those field measurements of known location, it would be crucial to show the real difference with digital line samples."*

As mentioned in line 355, the location of the field measurements is not known exactly, because the field measurements were originally recorded for other purposes, within other projects. The best we can do with the available data is to evaluate the bar-level agreement between (independent) field work and our digital line sampling approach. For the training and evaluation of the CNN we consider the digital line samples to be representative (line 344). We do agree that a comparison between digital line samples and geolocalized field samples would be interesting, but since this data is not available, we cannot provide it in this paper.

*"A better comparison with digital line samples should be carried out: I do not agree that ". . .overall, no bias exists between the field measurements and the digital line samples" (L 343-344; figure 6)."*

**Addressed in the paper: line 359-361, 363-365**

**Robust estimation**

Figure 11 shows the estimated grain size distribution for entire gravel bars in comparison to the ground truth data. Given appropriate training data, the model estimates the grain size distributions very accurately. Due to the inherent regularisation, deep learning models tend to be fairly robust.

**Manual component of the presented approach**

*"Comparison with human performance (section 5.4.4). Errors are not so small, see figure 15"*

The vertical axis label in Figure 15 seems to have been mangled during pdf generation. To clarify, the Y-axis in Figure 15 corresponds to the Dm, not the error. We compare the mean diameter (Dm) estimated by the CNN to the Dm from multiple annotations by different operators. This comparison requires a lot of manual labour (repeated, independent annotation by different people), thus only a small number of samples could be processed, which is not ideal for statistical analysis. However, it still gives a feeling for the human variation in the annotation process, with an average standard deviation across different operators of 0.5 cm for Dm. The max standard deviation from repeated annotation is 2.0 cm. We correct a small mistake in line 488: The standard deviation of the labels (0.5 cm) should be compared with the root mean squared error (RMSE) of the model predictions, not with the mean absolute error MAE.

**Presence of fine material**

We have already included two examples in Figure 8 b) and c) that show that the network can make robust predictions even in the case of slight disturbances caused by colmation through fine material, given such samples are provided during training.

**References**

Wohl, E. E., Anthony, D. J., Madsen, S. W., & Thompson, D. M. (1996). A comparison of surface sampling methods for coarse fluvial sediments. *Water Resources Research*, *32*(10), 3219-3226.

---

## Author Response (AR2)

**HESS GRAINet review answer**

Dear Matjaz Mikos,
We thank all the reviewers for their inputs. Please see our detailed answers below.
We hope that the reviewer's concerns could be solved with our clarification.

New changes in the manuscript are highlighted in red. Changes from the first revision are still highlighted in blue.

**Editor**

After receiving two reviews, there is a clear need for further revisions to take into account the suggestions of the both reviewers. Comments and suggestions of Referee #3 is a bit easier to answer - but please, follow the suggestions to add (American) literature - I am also aware of the Fehr method (Anastasi), but Bunte, Abt, ... should be used and mention.
We already refer to the proposed american literature in the introduction where appropriate (Bunte and Abt (2001), Rice and Church (2010)) . Since our data annotation strategy closely follows Fehr's line-sampling approach, we cannot avoid the reference to Fehr for obvious reasons. However, we now connect this surface sampling procedure to the commonly used terminology in the literature.
Our novel methodology has been evaluated on grain size data for Switzerland. Thus, we adopted Fehr, which is still considered the gold standard in Switzerland and used almost universally in Swiss engineering practice.
Nevertheless, we added an explanation in the paper (footnote line 79) that our methodology is **not** restricted to the Fehr sampling strategy. The proposed CNN can be trained to replicate any kind of grain size data, no matter how it has been sampled. May it be along a transect, a grid, or even a volume sample. (see also our previous answer to RC3).

I would also agree with his remark on mean grain diameter dm-values and its significance for the calibration of numerical sediment transport models. I would rather think into the direction of fraction models using several sediment fractions to be able to incorporate building of an armour layer and the influence of selective transport mechanism. An advanced method for grain size determination should overcome the limitations by a mean grain size d-m models. Why to apply a sophisticated method for d-m determination rather than to get a full GSD?
As we write in line: 91, the novelty of GRAINet is indeed its ability to predict the full grain size distribution at every location on a gravel bar. We agree that this offers a great potential to calibrate numerical models. However, we also show that our data-driven CNN is generic and can also be used to predict characteristic grain sizes. Aggregate statistics such as d_m have their role, too, e.g., it is straight-forward to visualize a characteristic grain size as a spatial map, whereas visualizing the full GSD in a spatially explicit manner is not trivial.

The comments from Referee#2 are more important. Please, give readers more details about the procedure and the equipment (UAV - we have applied Phantom DJI drones for rock fall applications and surface displacements on scree slopes), so that the reader would be aware about the possibilities. This is important, if such field campaigns are performed by

non-experts (we ask geodetical engineers that have pilot certificates). A further clarification is needed in this regard.

All the information needed to reproduce the UAV surveys was already given in the paper in section 3.1.

To cover the reviewer's concerns about the flying time, we added a paragraph in the discussion section 6.4 (lines: 641-647).

Please, follow the suggestion and redo the geographical cross validation. This is a critical issue to be able to generalise the results. Are you sure that grain size distributions at a series of gravel bars of the same river are not inter-related?

The grain sizes of the investigated gravel bars do not follow Sternberg's fluvial abrasion law, but are dominated by discontinuities. The grain size characteristics vary greatly along the same river (see Table C1). As we write on page 31 lines 584-593, the grain sizes may be impacted by several influences: changing river characteristics (slope, bed width), anthropogenic barriers (i.e., dams; these are not free-flowing rivers), tributaries (increased catchment areas), and artificial gravel replenishments.

We also explicitly discuss in line 575 that this experimental setup is valid for our specific dataset and that in different scenarios with slowly varying grain size properties, a river-based cross-validation may be needed to avoid overly optimistic results.

Our available dataset with 25 bars allows us to study the generalization across gravel bars (see title of section 5.4). In line 86 we already clarify the generalization experiment. The paper never claims that our method would generalize to unseen conditions. Rather, we explicitly mention in line 601 that the data-driven approach will only work if the training data and test data follow the same distribution, which goes beyond grain size distribution and also includes appearance (weather, lighting, surface cover of gravel, ...).

You should elaborate more on this issue (using classic papers from fluvial geomorphology on sediment sources and sediment links in a fluvial system). Please, compare results of the geographical cross validation using all data (this is already done) and only data from different rivers (skipping potentially cross-related data from neighbouring gravel bars on the same river). We should clarify this issue before going on with the publication process.

On the request of the reviewer, we have identified five bars that are per definition independent, as they belong to separated river reaches: *Aare km 040.8, Kl. Emme km 030.3, Emme -, Rhone km 083.3, Rhone km 114.0*.

These bars are separated from the other bars with the same river name by new inputs of sediment. Thus, for these bars, there are no neighbouring bars in the training data.

In our per-bar cross-validation experiment, the average error (MAE) for these five bars is 0.3cm higher than for random cross-validation, whereas the average error over **all** bars (including the supposedly not geographically separated ones) is 0.4cm higher. See the attached Figure 1 in this document. We conclude that, within our dataset, seeing bars from the same river during training does not lead to over-optimistic results.

Generalization is much more affected by unique local environmental factors (e.g. wet stones, algae covering) that were not seen during training. The dominant error case in the generalization experiment was that all samples with wet grains are from the same bar (i.e., *Aare km 156.7*; see also line 528 in the paper). We clearly see that *Aare km 040.8*, although on a separated river reach, performs better in our per-bar cross-validation than the majority of gravel bars along the river Aare. In general, we observe more pronounced drops for bars

with coarse grains (Gr. Entle, KL. Emme, Rhone), which we assume is due to sampling bias, as samples with coarse grain sizes are rare (see line 598, 636).

After these additional analyses we are firmly convinced that our study is correct (for the available dataset), and consistent with our careful discussion on the generalization performance in section 6.2. We would thus prefer to keep section "5.4 Generalization across gravel bars" and not replace it with a "per river name" cross-validation, with similar quantitative outcomes, but a smaller data basis. If the editor insists that the replacement is necessary, we would of course oblige. Nevertheless, we include this analysis in the paper section 6.4 (lines 577-583) with the new Figure in the Appendix Fig. H.

[Figure]

Figure 1: Increase of mean absolute error (MAE) from the random cross-validation to the cross-bar generalization experiment. A positive value means that the generalization experiment has a higher error.

**Report 1**

This revised manuscript has many improvements on issues like model architecture and training. Class activation maps also provide a very useful insight into how the model works. Overall, this method is very innovative and it produces results that are high quality and very difficult to achieve with other methods. However, the authors still do not clearly acknowledge the limitations of their method and this rests on 2 points: an unclear understanding of the logistical costs of acquiring sufficient data for GrainNet with a UAV and a false result for their so-called geographic cross validation.

First, the authors begin their response letter by stating that they do not see an issue with the acquisition of drone data at 0.25 cm of spatial resolution and characterise it as a 'minor technical detail'. I will therefore clarify my comment with a worked example. Start with a 1 hectare (100x100 metre) bar a a unit sampling area. The project uses a DJI P4 pro, let's simplify the problem by assuming a 90 degree FOV meaning that the image footprint is twice the flying altitude. The images were acquired at 16:9 aspect ratio with 5472x3648 pixels. From this we can derive that the drone was approximately 6.8 m above ground. Given that the method needs an orthomosaic, I will assume that the images are flown at 80% forward overlap with a 50% sidelap. The image height is 9.1 m. Between images the drone must move 20% of the image to get the 80% overlap. This is 1.82 meters. On the P4 pro and with the fastest SD card on the market, you need to leave abut 2s for the mage to write to disk, anything less and the drone will start missing images during the mission. So the optimal flight method is to get a slow continuous motion of the drone. 1.82 meters in 2s is 0.9 m/s. It will therefore take roughly 0.9 minutes to complete 1 line of 100m. With the images being 12m wide and a 50% sidelap, We need about 17 flight lines to cover 1 hectare. For a total flight time of about 15 minutes/hectare.

Now consider an alternative setup that is used to get grain size data for alternative texture mapping methods. In this case imagery acquired at 2-3 cm of spatial resolution is suitable. In this case, flying a P4 pro at 50 m altitude will deliver suitable imagery at about 2cm. At 50m, the footprint of 1 image is 100m x 56m. At 80 overlap and the same 2s interval between images, the drone flies at 5.6 m/s. Given the image width, we only need 2 lines to cover the hectare. Meaning that the total operation needs only 34 seconds/hectare.

Therefore, data acquisition for GrainNet requires drone operations that are 30x longer than for older methods. That is not trivial and readers deserve to know this fact.

The authors' suggestion that magnification can solve the problem is incorrect. When you magnify you increase the focal length and thus reduce the image footprint, flight velocity for SfM acquisition remains the same. Whilst it is true that a higher resolution camera could indeed improve things, that trend is very slow. The current UAV market for science is now dominated by consumer, non-scientific, drones made by DJI. The simple reason is cost. The P4 pro resolution of 20 Mpix is already on the high side. The only way to improve the performance would be to use top of the line cameras that have high speed writing buses. For example, mounting a Canon EOS on a big drone like a Matrice 600 would indeed be much faster, but then you are talking of a 1 order magnitude increase in cost for drone equipment. Either way, the acquisition of appropriate data for GrainNet is a significant barrier to access.

To cover the reviewer's concerns about the flying time, we added a paragraph in the discussion section 6.4 to make the reader aware of this fact. We note that all drone data acquisition was performed by a hydrological consulting SME in the course of their commercial operations, which shows that the approach, even in its current, unoptimised form, is fairly practical and accessible.

The second issue is the geographic cross validation. My view is still that the authors' approach is mistaken and unjustified in geomorphology literature. The authors state on line 564 that there is no strong correlation between grain sizes on the same rivers. This statement is not evidenced and it flies in the face of decades of fluvial geomorphology. It has long been known that grain size decreases exponentially with distance downstream with periodic discontinuities (Rice, 1999; Rice and Church, 1998, 2001). This was again observed in recent remote sensing studies(Carbonneau et al., 2005). So barring the incidence of a source of coarse grained material, two successive bars on the same river can be expected to have a similar grain size composed of similar material of the same source. So unless the authors can show that between each and every one of their sampling bars there is a new input of sediment, then we must expect that the majority of neighbouring bars in the dataset are similar and LOOCV is not an appropriate method. I again make the request that the authors revise this process to hold-out entire rivers.

Please see the answer to the editor above. Our dataset includes five cases where only a single bar is located on the same river reach and thus is independent of all others bars with the same river name. We looked at the cross-validation results for these bars, which are no worse than for other bars that are not formally independent (in the sense of "separated by significant sediment inputs"). The analysis confirms our previous argument that grain sizes along the investigated Swiss rivers are dominated by anthropogenic influences (dams, channels, replenishments) as well as by frequent tributaries. Correlations due to continuous abrasion are minimal.

This is critical because as it stands, this method does provide unprecedented data over a gravel surface, but as I show above, the logistic costs of data acquisition are an order of magnitude more in time or cost when compared to older methods. If it turns out that the method does not generalise to new rivers, then local calibration will be needed at each acquisition thus increasing the total cost of the method. I do not doubt that in certain applications, such a large field effort will be justified in order to produce such high quality outputs, but the reader deserves to get a clear indication of these costs upfront.

Patrice Carbonneau
December 2020

We hope that the reviewer's concerns could be solved with our clarification.
The paper never claims that the presented method would generalize to unseen conditions. Rather, we explicitly mention in line 601 that such a data-driven approach will only work if the training data and test data follow the same data distribution, which goes beyond grain size distribution, but also includes appearance (weather, lighting, environmental factors).

**Report 2**

Overall, I can see several improvements in this revised version of the manuscript. That said, I still have some concerns about this manuscript:

1.      Introduction. I see small improvements in this Section. I think that the main points (suggests) in my previous review were not well addressed. For simplicity such points are reported below (lines refer to the previous version of the manuscript):

L 15-28. This part is not very useful. It would be more useful to focus on why grain size data are crucial (e.g. process understanding, modelling).
Done. We now have included the motivation to collect grain size data to advance the understanding of stream processes with references to Bunte and Abt (2001).
L 38-42. Reference to traditional approaches is very poor. I would avoid the reference to Fehr (1987), maybe a good reference in the German-speaking countries but not worldwide (and in an international journal). I would suggest to look and refer to classical works by Church, Bunte, and many others. For instance, a look to Bunte and Abt (2001, USDA) would be very useful to put this work in the general context of sediment sampling in gravel-bed rivers.
Done
L 56. "….is more efficient than traditional field measurements…": I would say that automatic grain size is much less time consuming but it is also, commonly, less accurate. This should be pointed out since it is probably not obvious for readers who are not familiar with sediment sampling.
Done
2.      Testing of the approach. L 606-607. "…Our CNN-based approach makes it possible to robustly estimate grain size distributions and characteristic mean diameters from raw images…". Comparison with field data (real data) is weak in this work.
We did our best with the available field data by comparing at the bar level. Precise geolocation was not available for the field data, which was assembled from multiple past projects and campaigns (in operational practice). We agree that with today's technology (e.g., real-time GNSS) it may be useful to routinely record precise geolocation, and have mentioned this possibility in the paper (line 374). But the data that was available to us, collected with today's field sampling practices, did not include precise georeference.

Overall, the authors do not fully recognize that field sampling is more accurate that sampling from images. It seems to me that the final message of this work is as follows: sediment characterization from images is more reliable and accurate than characterization by field measurements. I do not think this is the case: it would be useful to clarify better advantages and limitations of both approaches (this would be very useful in the "Introduction").
We clarified advantages and limitations in line 50. However, our message is certainly not that sampling from images is more accurate and reliable, and we never say this in the paper. If we claim any advantage over the gold standard of field measurements, then it is the ability to scale up the data acquisition to advance the calibration of automatic data-driven methods that ultimately lead to holistic analyses of gravel bars (e.g. dense maps Sec. 5.3.5, Fig. 17).

Finally, I think that a sound test to assess the performance of the model was not carried out in this work: it should be relevant to point out this for future research development.
We agree that a more extended comparison of the manual digital line sampling and field samples should be done in the future. (see line 374). This would require precisely geolocated line samples in the field that could then be compared against manual digital line samples. Nevertheless, in our view the extensive evaluation of the model against the digital line samples clearly demonstrates the advantages and limitations of this approach.

SPECIFIC COMMENTS

L 480-481. "…In order to calibrate numerical bedload transport models, a single representative dm-value of a gravel bar or a cross-section is essential…": I am not so sure about this, could you better support and justify this statement?
We removed this sentence as it was misleading. We agree with the opinion that the potential for numerical model calibration lies in the ability of GRAINet to estimate the full grain size distributions at a high spatial resolution. Nevertheless, robust estimates of characteristic grain sizes at the bar level are important to support large scale analysis along gravel-bed rivers.

---

## Author Response (AR3)

Please find attached the final manuscript.

We thank the editor and all reviewers for their contribution and suggestions to clarify our work.

Best regards
Nico Lang